# Differentiable Gaussianization Layers for Inverse Problems Regularized by Deep Generative Models

**Dongzhuo Li**
ExxonMobil Technology & Engineering Company
`dongzhuo.li@exxonmobil.com`

## Abstract

Deep generative models such as GANs, normalizing flows, and diffusion models are powerful regularizers for inverse problems. They exhibit great potential for helping reduce ill-posedness and attain high-quality results. However, the latent tensors of such deep generative models can fall out of the desired high-dimensional standard Gaussian distribution during inversion, particularly in the presence of data noise and inaccurate forward models, leading to low-fidelity solutions. To address this issue, we propose to reparameterize and Gaussianize the latent tensors using novel differentiable data-dependent layers wherein custom operators are defined by solving optimization problems. These proposed layers constrain inverse problems to obtain high-fidelity in-distribution solutions. We validate our technique on three inversion tasks: compressive-sensing MRI, image deblurring, and eikonal tomography (a nonlinear PDE-constrained inverse problem) using two representative deep generative models: StyleGAN2 and Glow. Our approach achieves state-of-the-art performance in terms of accuracy and consistency.

## 1 Introduction

Inverse problems play a crucial role in many scientific fields and everyday applications. For example, astrophysicists use radio electromagnetic data to image galaxies and black holes (Högbom, 1974; Akiyama et al., 2019). Geoscientists rely on seismic recordings to reveal the internal structures of Earth (Tarantola, 1984; Tromp et al., 2005; Virieux & Operto, 2009). Biomedical engineers and doctors use X-ray projections, ultrasound measurements, and magnetic resonance data to reconstruct images of human tissues and organs (Lauterbur, 1973; Gemmeke & Ruiter, 2007; Lustig et al., 2007). Therefore, developing effective solutions for inverse problems is of great importance in advancing scientific endeavors and improving our daily lives.

Solving an inverse problem starts with the definition of a forward mapping from parameters $\mathbf{m}$ to data $\mathbf{d}$, which we formally write as

$$\mathbf{d} = f(\mathbf{m}) + \boldsymbol{\epsilon}, \tag{1}$$

where $f$ stands for a forward model that usually describes some physical process, $\boldsymbol{\epsilon}$ denotes noise, $\mathbf{d}$ the observed data, and $\mathbf{m}$ the parameters to be estimated. The forward model can be either linear or nonlinear and either explicit or implicitly defined by solving partial differential equations (PDEs). This study considers three representative inverse problems: **Compressive Sensing MRI**, **Deblurring**, and **Eikonal (traveltime) Tomography**, which have important applications in medical science, geoscience, and astronomy. The details of each problem and its forward model are in App. A.

The forward problem maps $\mathbf{m}$ to $\mathbf{d}$, while the inverse problem estimates $\mathbf{m}$ given $\mathbf{d}$. Unfortunately, inverse problems are generally under-determined with infinitely many compatible solutions and intrinsically ill-posed because of the nature of the physical system. Worse still, the observed data are usually noisy, and the assumed forward model might be inaccurate, exacerbating the ill-posedness. These challenges require using regularization to inject *a priori* knowledge into inversion processes to obtain plausible and high-fidelity results. Therefore, an inverse problem is usually posed as an optimization problem:

$$\arg\min_{\mathbf{m}} \; (1/2) \left\| \mathbf{d} - f(\mathbf{m}) \right\|_2^2 + \mathcal{R}(\mathbf{m}), \tag{2}$$

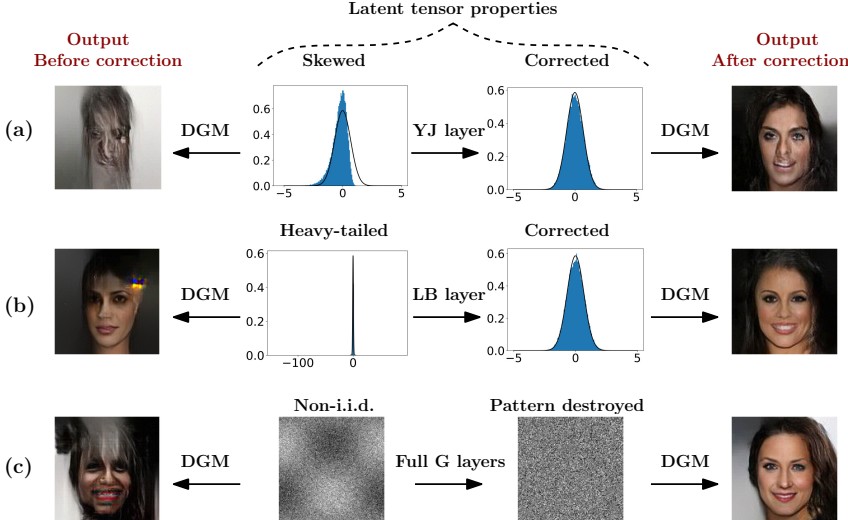

Figure 1: Comparison of images generated by a deep generative model (DGM), Glow, using latent tensors that deviate from a spherical Gaussian distribution (left) and those after corresponding corrections (right). The visual effects highlight the necessity of keeping the latent tensor within such a distribution during inversion. The second column shows the characteristics of deviated latent tensors: (a) histogram: i.i.d. components but the distribution is skewed; (b) histogram: i.i.d. components but the distribution is heavy-tailed; (c) latent tensor image: non-i.i.d. entries. The first column shows the corresponding outputs of a Glow network. The third column shows latent tensors corrected by (a) the Yeo-Johnson layer (YJ), (b) the Lambert $W \times F_X$ layer (LB), and (c) the full set of our Gaussianization layers (G layers). Those corrected latent tensors map to realistic images shown in the fourth column. All latent tensors have a norm of $0.7\sqrt{\text{vec dim}}$ because of the Gaussian Annulus Theorem (App. I) and the fact that Glow works best with a temperature smaller than one (see Fig. 9). Additional examples for **StyleGAN2** and **Stable Diffusion** (Rombach et al., 2022) can be found in App. B.

where $\mathcal{R}(\mathbf{m})$ is the regularization term. Beyond traditional regularization methods such as the Tikhonov regularization and Total Variation (TV) regularization, deep generative models (DGM), such as VAEs (Kingma & Welling, 2013), GANs (Goodfellow et al., 2014), and normalizing flows (Dinh et al., 2014; 2016; Kingma et al., 2016; Papamakarios et al., 2017; Marinescu et al., 2020), have shown great potential for regularizing inverse problems (Bora et al., 2017; Van Veen et al., 2018; Hand et al., 2018; Ongie et al., 2020; Asim et al., 2020; Mosser et al., 2020; Li et al., 2021; Siahkoohi et al., 2021; Whang et al., 2021; Cheng et al., 2022; Daras et al., 2021; 2022). Such deep generative models directly learn from training data distributions and are a powerful and versatile prior. They map latent vectors $\mathbf{z}$ to outputs $\mathbf{m}$ distributed according to an *a priori* distribution: $\mathbf{m} = g(\mathbf{z}) \sim p_{\texttt{target}}, \mathbf{z} \sim \mathcal{N}(\mathbf{0}, \mathbf{I})$, for example. The framework of DGM-regularized inversion (Bora et al., 2017) is

$$\arg\min_{\mathbf{z}} \; (1/2) \left\| \mathbf{d} - f \circ g(\mathbf{z}) \right\|_2^2 + \mathcal{R}'(\mathbf{z}), \tag{3}$$

where the deep generative model $g$, whose layers are frozen, reparameterizes the original variable $\mathbf{m}$, acting as a hard constraint. Instead of optimizing for $\mathbf{m}$, we now estimate the latent variable $\mathbf{z}$ and retrieve the inverted $\mathbf{m}$ by forward mappings. Since the latent distribution is usually a standard Gaussian, the new (optional) regularization term $\mathcal{R}'(\mathbf{z})$ can be chosen as $\beta \|\mathbf{z}\|_2^2$ for GANs and VAEs, where $\beta$ is a weighting factor. See App. J.1 for more details on a similar formulation for normalizing flows. Since the optimal $\beta$ depends on the problem and data, tuning $\beta$ is highly subjective and costly.

However, this formulation of DGM-regularized inversion still leads to unsatisfactory results if the data are noisy or the forward model is inaccurate, as shown in Fig. 20, even if we fine-tune the weighting parameter $\beta$.

To analyze this problem, first recall that a well-trained DGM has a latent space (usually) defined on a standard Gaussian distribution. In other words, a DGM either only sees standard Gaussian latent

tensors during training (*e.g.*, GANs) or learns to establish 1-1 mappings between training examples and typical samples from the standard Gaussian distribution (*e.g.*, normalizing flows, App. J.2). As a result, the generator may map out-of-distribution latent vectors to unrealistic results. We show in Fig. 1 the visual effects of several types of deviations of latent vectors from a spherical Gaussian (with a temperature of 0.7). It can be seen that (1) the latent tensor should have i.i.d. entries, and (2) these entries should be distributed as a 1D standard Gaussian in order to generate plausible images. Since the traditional DGM-regularized inversion lacks such Gaussianity constraint, we conjecture[1] that the latent tensor deviates from the desired high-dimensional standard Gaussian distribution during inversion, leading to poor results. Our observations and reasoning motivate us to propose a set of differentiable Gaussianization layers to reparameterize and Gaussianize the latent vectors of deep generative models (*e.g.*, StyleGAN2 (Karras et al., 2020) and Glow (Kingma & Dhariwal, 2018)) for inverse problems. The implementation is available *here*.

## 2 METHOD

### 2.1 OVERVIEW - REPARAMETERIZATION USING GAUSSIANIZATION LAYERS

Our solution is based on a necessary condition of the standard Gaussian prior on latent tensors. Let us define a *partition* of a latent tensor $\mathbf{z} \in \mathbb{R}^n$ as the collection of non-overlapping patches $P(\mathbf{z}) = \{\mathbf{z}_i\}_{i=1,\cdots,N}$, where the patches $\mathbf{z}_i \in \mathbb{R}^D$ are of the same dimension and can be assembled as $\mathbf{z}$, *i.e.*, $n = N \times D$. If the latent tensor $\mathbf{z} \sim \mathcal{N}(\mathbf{0}, \mathbf{I})$, then for all $\mathbf{z}_i$ from any partition $P(\mathbf{z})$, we have $\mathbf{z}_i \sim \mathcal{N}(\mathbf{0}, \mathbf{I})$.

Note that $\mathbf{z}$ is a symbolic representation of the latent tensor. In a specific DGM, $\mathbf{z}$ can be either a 2D/3D tensor or a list of such tensors corresponding to a multi-scale architecture (App. D). Even though there are numerous partition schemes, such as random grouping of components, we choose the simplest: partitioning $\mathbf{z}$ based on neighboring components, which works well in practice.

To constrain $\mathbf{z}_i \sim \mathcal{N}(\mathbf{0}, \mathbf{I})$ during inversion, we reparameterize $\mathbf{z}_i$ by constructing a mapping $h : \mathbf{v}_i \to \mathbf{z}_i$, such that $\mathbf{z}_i \sim \mathcal{N}(\mathbf{0}, \mathbf{I})$. The new variables $\mathbf{v}_i$ are of the same dimension as $\mathbf{z}_i$. Suppose that we have constructed the patch-level mapping $h$, we can obtain a mapping $h^\dagger$ at the tensor level, such that $\mathbf{z} = h^\dagger(\mathbf{v})$, where $\mathbf{v}$ is assembled from $\mathbf{v}_i$ in the same way as $\mathbf{z}$ from $\mathbf{z}_i$. For example, $h^\dagger = \mathtt{diag}(h, \cdots, h)$ if patches are extracted from neighboring components and are concatenated into a vectorized $\mathbf{v}$. Hence, the original DGM-regularized inversion 3 becomes

$$\underset{\mathbf{v}}{\arg\min}\ (1/2) \left\| \mathbf{d} - f \circ g \circ h^\dagger(\mathbf{v}) \right\|_2^2. \tag{4}$$

Since we are imposing a constraint through reparameterization, there is no need to include the regularization term $\mathcal{R}'(\mathbf{z})$. The new formulation is still an unconstrained optimization problem, enabling us to use highly efficient unconstrained optimizers, such as L-BFGS (Nocedal & Wright, 2006) and ADAM (Kingma & Ba, 2015).

The remaining critical piece is to construct $h$, and it leads to our Gaussianization layers. First, we translate the constraint of $\mathbf{z}_i = h(\mathbf{v}_i) \sim \mathcal{N}(\mathbf{0}, \mathbf{I})$ into the following optimization problem:

$$\underset{h}{\arg\min}\ D_{\mathrm{KL}}\left( p_Z\left(h\left(\mathbf{v}_i\right)\right) \| \mathcal{N}(\mathbf{0}, \mathbf{I}) \right), \tag{5}$$

where $p_Z$ is the probability density function (PDF) of $\mathbf{z}_i$. Second, we adopt the framework proposed by precursor works of normalizing flows on Gaussianization (Chen & Gopinath, 2000; Laparra et al., 2011) to solve this optimization problem. The KL-divergence can be decomposed as the sum of the multi-information $I(\mathbf{z}_i)$ and the marginal negentropy $J_m(\mathbf{z}_i)$ (Chen & Gopinath, 2000; Meng et al., 2020):

$$D_{\mathrm{KL}}\left( p_Z\left(\mathbf{z}_i\right) \| \mathcal{N}(\mathbf{0}, \mathbf{I}) \right) = I(\mathbf{z}_i) + J_m(\mathbf{z}_i), \tag{6}$$

where

$$I(\mathbf{z}_i) = D_{\mathrm{KL}}\left( p_Z\left(\mathbf{z}_i\right) \left\| \prod_j^D p_j(z_i^{(j)}) \right. \right), \text{ and } J_m(\mathbf{z}_i) = \sum_{j=1}^D D_{\mathrm{KL}}\left( p_j(z_i^{(j)}) \middle\| \mathcal{N}(0, 1) \right). \tag{7}$$

---

[1]Out-of-distribution/typicality tests are very challenging for high-dimensional data (Rabanser et al., 2019; Nalisnick et al., 2019). Also, since latent tensors are Gaussian/Gaussian-like, it is hard to conduct dimension reduction for such tests.

Here $z_i^{(j)}$ denotes the $j$-th component of patch vector $\mathbf{z}_i$, and $p_j$ stands for the marginal PDF for that component. The multi-information $I(\mathbf{z}_i)$ quantifies the independence of the components of $\mathbf{z}_i$, while the marginal negentropy $J_m(\mathbf{z}_i)$ describes how close each component is to a 1D standard Gaussian. With this decomposition, the optimization procedure depends on the facts: (1) the KL divergence and a standard Gaussian in Eq. 6 are invariant to an orthogonal transformation, and (2) the multi-information term is invariant to a component-wise invertible differentiable transformation (App. J.3). As a result, we perform Gaussianization in two steps:

1. Minimize the multi-information $I(\mathbf{z}_i)$. This is done by an orthogonal transformation that keeps the overall KL divergence the same but increases the negentropy $J_m(\mathbf{z}_i)$. We achieve this by using our independent component analysis (ICA) layer. ICA is the optimal choice since it maximizes non-Gaussianity so that the subsequent marginal Gaussianization step removes it and results in a large decrease in $D_{\mathrm{KL}}\left(p_Z\left(h\left(\mathbf{v}_i\right)\right) \| \mathcal{N}(\mathbf{0}, \mathbf{I})\right)$.

2. Minimize the marginal negentropy $J_m(\mathbf{z}_i)$ by component-wise operations that perform 1D Gaussianization of marginal distributions $p_{j,\,j=1,\cdots,D}$. The multi-information does not change under component-wise invertible operations. Therefore, the overall KL divergence between $\mathbf{z}_i$ and the Gaussian distribution decreases.

The Gaussianization steps are well-aligned with the motivating example (Fig. 1). To constrain DGM outputs to be plausible, one should make components within latent tensor patches independent (or destroy the patterns) (Fig. 1(c)) and shape the 1D distribution as close as possible to Gaussian (Fig. 1(a)(b)).

We will see that $h$ is parameterized by an orthogonal matrix and two scalar parameters in 1D Gaussianization layers. Unlike conventional neural network layers, the input-data-dependent Gaussianization layers are not defined by learning from a dataset ahead of time but by solving certain optimization problems on the job (Fig. 18). Special care should be taken to implement the gradient computation correctly and ensure that they pass the finite-difference convergence test (App. G). As an overview, the composition of our proposed layers is:

$$\mathbf{v} \rightarrow \textbf{Whitening} \rightarrow \textbf{ICA} \rightarrow \textbf{Yeo-Johnson} \rightarrow \textbf{Lambert } W \times F_X \rightarrow \textbf{Standardization} \rightarrow \mathbf{z},$$

where whitening and ICA belong to the first step and the rest belong to the second step. We will discuss in App. K some possible simplifications of the layers in practice after our ablation studies.

The overall proposed inversion process (one iteration) is illustrated in Fig. 2.

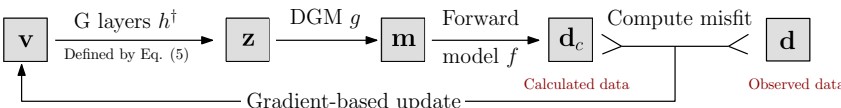

Figure 2: Illustration of the proposed inversion process. Gradient computation in the Gaussianization layers is enabled by the implicit function theorem and automatic differentiation (App. G). We use the L-BFGS optimizer (Nocedal & Wright, 2006) to update $\mathbf{v}$.

## 2.2 REDUCING MULTI-INFORMATION – ICA LAYER

The orthogonal matrix $\boldsymbol{W}$ is constructed by the independent component analysis (ICA). The input is the patch vectors $\{\mathbf{v}_i|_{i=1,\cdots,N}\}$. The ICA algorithm first computes the input-dependent orthogonal matrix $\boldsymbol{W}$ and then computes $\mathbf{p}_i = \boldsymbol{W}^\top \mathbf{v}_i,\,_{i=1,\cdots,N}$ as the output. The orthogonal matrix $\boldsymbol{W}$ makes the entries of each $\mathbf{p}_i$ independent random variables.

We use the FastICA algorithm (Hyvarinen, 1999; Hyvärinen & Oja, 2000), which employs a fixed-point algorithm to maximize a contrast function $\Phi$ (*e.g.*, the logcosh function), for our ICA layer to reduce multi-information.

The FastICA algorithm typically requires that the data are pre-whitened. We adopt the ZCA whitening method or the iterative whitening method introduced in Hyvarinen (1999) (App. E). With the whitened data, we compute $\boldsymbol{W}$ using a damped fixed-point iteration scheme:

$$\boldsymbol{W} = \frac{1}{N}\left[\alpha \boldsymbol{V} \phi\left(\boldsymbol{W}^\top \boldsymbol{V}\right)^\top - \boldsymbol{W} \mathtt{diag}\left(\phi'\left(\boldsymbol{W}^\top \boldsymbol{V}\right)\mathbf{1}\right)\right], \tag{8}$$

where $\mathbf{1}$ is an all-one vector, the column vectors of $\boldsymbol{V}$ are $\{\mathbf{v}_i|_{i=1,\cdots,N}\}$, $\phi(\cdot) = \Phi'(\cdot)$, $\alpha \in (0,1)$, and we use $\alpha = 0.8$ throughout our experiments. To save computation time, we only perform a maximum of 10 iterations. The details of the whole algorithm can be found in App. E.

We set the initial $\boldsymbol{W}$ as an identity matrix. If the input vectors are already standard Gaussian (in-distribution), the computed $\boldsymbol{W}$ will still be an identity matrix, which maps the input to the same output. In practice, the empirical distribution from finite samples is not a standard Gaussian, so $\boldsymbol{W}$ is not an identity matrix but another orthogonal matrix, which still maps standard Gaussian input vectors to standard Gaussian vectors as output. For this reason, we sample starting $\{\mathbf{v}_i|_{i=1,\cdots,N}\}$ from the standard Gaussian distribution to start inversions with plausible outputs.

## 2.3 REDUCING MARGINAL NEGENTROPY

For 1D Gaussianization, we choose a combination of the Yeo-Johnson transformation that reduces skewness and the Lambert $W \times F_X$ transformation that reduces heavy-tailedness. Both are layers based on optimization problems with only one parameter, which is cheap to compute and is easy to back-propagate the gradient. Eq. 7 requires us to perform such 1D transformations for each component of the random vectors. In other words, we need to solve the same optimization problem for $D$ times, which imposes a substantial computational burden. Instead, we empirically find it acceptable to share the same optimization-generated parameter across all components. In other words, we perform only a single 1D Gaussianization, treating all entry values in the latent vector as the data simultaneously.

**Power transformation layer** We propose to use the power transformation or Yeo-Johnson transformation (Yeo & Johnson, 2000) to reduce the skewness of distributions. As shown in Fig. 3(a), the form of the Yeo-Johnson activation function depends on parameter $\lambda$. If $\lambda = 1$, the mapping is an identity mapping. If $\lambda > 1$, the activation function is convex, compressing the left tail and extending the right tail, reducing the left-skewness. If $\lambda < 1$, the activation function is concave, which oppositely reduces the right-skewness. We refer the readers to App. E for details.

**Lambert $W \times F_X$ layer** Due to noise and inaccurate forward models, we observe that the distribution of latent vector values tends to be shaped as a heavy-tailed distribution during the inversion process. To reduce the heavy-tailedness, we adopt the Lambert $W \times F_X$ method detailed in Goerg (2015).

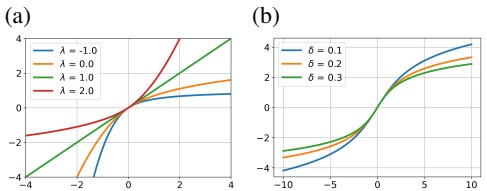

We use the parameterized Lambert $W \times F_X$ distribution family to approximate a heavy-tailed input and solve an optimization to estimate an optimal parameter $\delta$ (App. E), with which the inverse transformation maps the heavy-tailed distribution towards a Gaussian.

Figure 3: The nonlinear activation functions from (a) the power transformation (Yeo-Johnson) layer and (b) the Lambert $W \times F_X$ layer.

Fig. 3(b) shows that the Lambert $W \times F_X$ layer acts as a nonlinear squashing function. As $\delta$ increases, it compresses more the large values and reduces the heavy-tailedness. Intuitively, the Lambert $W \times F_X$ layer can also be interpreted as an intelligent way of imposing constraints on the range of values instead of a simple box constraint. We refer the readers to App. E for more details about the optimization problem and implementation.

**Standardization with temperature** Since the Lambert $W \times F_X$ layer output may not necessarily have a zero mean and a unit (or a prescribed) variance, we standardize the output using

$$\boldsymbol{z} = (\boldsymbol{x} - \mathbb{E}[\boldsymbol{x}])/\sqrt{\mathrm{Var}(\boldsymbol{x})} * \gamma, \tag{9}$$

where $\gamma$ is the temperature parameter suggested in Kingma & Dhariwal (2018).

## 3 RELATED WORK

**End-to-end NNs for inverse problems**   There are numerous end-to-end neural networks designed for inverse problems, using CNNs (Chen et al., 2017; Jin et al., 2017; Sriram et al., 2020), GANs (Mardani et al., 2018; Lugmayr et al., 2020; Wang et al., 2018), invertible networks (Ardizzone et al., 2018), and diffusion models (Kawar et al., 2022). The general idea is simple: train a neural network that directly maps observed data to estimated parameters. Even though such methods seem effective in a few applications, DGM-regularized inversion with our Gaussianization layers is preferable for the following reasons. First, the forward modeling can be so expensive computationally that it is infeasible to collect a decent training datasets for some applications. For example, one large-scale fluid mechanics or wave propagation simulation can take hours, if not days. Second, the relationship between parameters and data can be highly nonlinear, and multiple solutions may exist. An end-to-end network may map data to interpolated solutions that are not realistic. In comparison, our method can start from different initializations or even employ sampling techniques to address this issue. Third, the configuration of data collection can change from experiment to experiment. It is impractical, for example, to re-train the network each time we change the number and locations of sensors. In contrast, not only can our method deal with this situation, but it can even use the same DGM for different forward models, as we can see in the compressive sensing MRI and eikonal tomography examples. While almost all end-to-end methods are only applied to linear inverse problems, our Gaussianization layers are also effective in nonlinear problems.

**Other techniques to improve DGM-regularized inversion**   In high-dimensional space, the probability mass of a standard Gaussian distribution concentrates within the so-called Gaussian typical set (App. I). To be in the Gaussian typical set, one necessary but not sufficient condition is to be within an annulus area around a high-dimensional sphere (App. I). Utilizing this geometric property, DGM-regularized inversion methods like Bojanowski et al. (2017) and Liang et al. (2021) force updated latent vectors to stay on the sphere. This strategy is closely related to spherical interpolation (White, 2016). We call this strategy the **spherical constraint** for inversion. In the original StyleGAN2 paper, the authors also noticed that in image projection tasks, the noise maps tend to have leakage from signals (Karras et al., 2020) – the same phenomenon we discussed. They proposed a multi-scale noise regularization term (**NoiseRlg**) to penalize spatial correlation in noise maps. We extend the same technique to our inverse problems for comparison. Note that we use a whitening layer before the ICA layer. The **whitening layer** can be used alone, similar to Huang et al. (2018) and Siarohin et al. (2018), whose performance will be reported in ablation studies. Also, Wulff & Torralba (2020) observed that for StyleGAN2, the Leaky ReLU function can "Gaussianize" latent vectors in the W space. **CSGM-w** (Kelkar & Anastasio, 2021) utilizes this *a priori* knowledge to improve DGM-regularized compressive sensing problems. To further compare with the Gaussianizaion layers, we in addition propose an alternative idea that reparameterizes latent vectors using learnable orthogonal matrices (Cayley parameterization) and fixed latent vectors, which is closely related to the work of orthogonal over-parameterized training (Liu et al., 2021) (App. H ). Recently, score-based generative models have started to show promise for solving inverse problems (Jalal et al., 2021; Song et al., 2021). However, they have been mainly applied to linear inverse problems and challenged by noisy data (Kawar et al., 2021). Besides, scored-based methods might not work for certain physics-based inverse problems, since Gaussian noise parameters may break the physics simulation engine[2].

## 4 EXPERIMENTS

We consider three representative inversion problems for testing: compressive sensing MRI, image deblurring, and eikonal traveltime tomography. For MRI and eikonal tomography, we used synthetic brain images as inversion targets and used the pre-trained StyleGAN2 weights from Kelkar & Anastasio (2021) (trained on data from the databases of fastMRI (Zbontar et al., 2018; Knoll et al., 2020), TCIA-GBM (Scarpace et al., 2016), and OASIS-3 (LaMontagne et al., 2019)) for regularization. We used the test split of the CelebA-HQ dataset (Karras et al., 2018) for deblurring, and the DGM is a Glow network trained on the training split. We refer readers to App. F for details on datasets and training.

---

[2]For example, in elastic waveform inversion, initializing material properties using Gaussian noise may create unrealistic scenarios where the P-wave speed is lower than the S-wave speed at some spatial locations.

We tested each parameter configuration in each inversion on 100 images (25 in the eikonal tomography due to its expensive forward modeling). Since the deep generative models are highly nonlinear, the results may get stuck in local minima. Thus, we started inversion using three different randomly initialized latent tensors for each of the 100 or 25 images, picked the best value among the three for each metric, and reported the mean and standard deviation of those metrics, except for CSGM-w, TV, and NoiseRlg, where the initialization is fixed. The metrics we used were PSNR, SSIM (Wang et al., 2004), and an additional LPIPS (Zhang et al., 2018) for the CelebA-HQ data. We used the LBFGS (Nocedal & Wright, 2006) optimizer in all experiments except TV, noise regularization, and CSGM-w, which use FISTA (Beck & Teboulle, 2009) or ADAM (Kingma & Ba, 2015). The temperature was set to 1.0 for StyleGAN2 and 0.7 for Glow.

## 4.1 COMPRESSIVE SENSING MRI USING STYLEGAN2

The mathematical model of compressive sensing MRI is

$$\mathbf{d} = \boldsymbol{A}\mathbf{m} + \boldsymbol{\epsilon}, \tag{10}$$

where $\boldsymbol{A} \in \mathbb{C}^{M \times N}$ is the sensing matrix, which consists of FFT and subsampling in the k-space (frequency domain). Eq. 10 is an under-determined system, and we use Accl $= N/M$ to denote the acceleration ratio. We also added i.i.d. complex Gaussian noise with a signal-to-noise ratio (SNR) of 20 dB or 10 dB to the measured data. See App. A.1 for more background information.

Table 1 compares the results from total variation regularization (TV), noise regularization (NoiseRlg) (Karras et al., 2020), spherical constraint/reparameterization: $\boldsymbol{z} = \boldsymbol{v}/\|\boldsymbol{v}\|_2 * \sqrt{\texttt{dim}(\boldsymbol{v})}$, CSGM-w (Kelkar & Anastasio, 2021), our proposed orthogonal reparameterization (Orthogonal), and our proposed Gaussianization layers (G layers). Fig. 4 shows examples of inversion results. In the base case where Accl=8x and SNR=20 dB, the Gaussianization layers gives the best scores, and this advantage gets more significant when data SNR decreases to 10 dB. Interestingly, the scores from all methods improve significantly if we make the system better determined (*i.e.*, Accl=2x), and the performance of TV, spherical constraint, and Gaussianization layers become more similar in this scenario. We conclude that our proposed Gaussianization layers are effective and more robust than other methods, particularly in low-SNR scenarios.

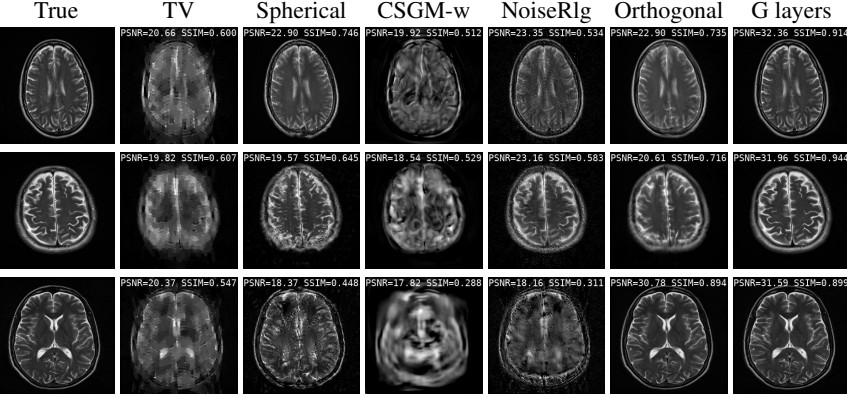

Figure 4: Comparison of compressive sensing MRI inversion results (Accl=8x, SNR=20 dB).

**Ablation study** Table 2 summarizes the ablation study on the components of the Gaussianization layers. We kept the standardization layer on for all cases. The conclusions are as follows: 1. The ICA layer plays the most significant role in improving result scores; 2. The whitening/ZCA layer alone is not effective; 3. The Yeo-Johnson (YJ) and the Lambert $W \times F_X$ (Lambt) layers are more effective when the noise level is higher (*e.g.*, SNR=10 dB vs. 20 dB). Their performance seems data-dependent: when SNR=20 dB, YJ seems more effective, while Lambt gives the best scores when SNR=10 dB. But the overall improvement from these two layers is marginal. In practice, one may only use ICA and one of the 1D Gaussianization layers. Additionally, we tested the effect of patch size on the z (style) vectors (App. C). We find that the largest possible patch size gives the best results.

Table 1: Comparison of compressive sensing MRI results.

| Method | Accl = 8x, SNR = 20 dB | | Accl = 8x, SNR = 10 dB | | Accl = 2x, SNR = 20 dB | |
| --- | --- | --- | --- | --- | --- | --- |
| | PSNR↑ | SSIM↑ | PSNR↑ | SSIM↑ | PSNR↑ | SSIM↑ |
| TV | $20.20_{\pm 1.33}$ | $0.60_{\pm 0.063}$ | $19.78_{\pm 1.32}$ | $0.57_{\pm 0.058}$ | $32.92_{\pm 1.36}$ | $0.91_{\pm 0.024}$ |
| Spherical | $26.93_{\pm 6.57}$ | $0.79_{\pm 0.159}$ | $22.78_{\pm 5.38}$ | $0.61_{\pm 0.246}$ | $32.90_{\pm 3.50}$ | $0.90_{\pm 0.061}$ |
| CSGM-w | $20.19_{\pm 2.11}$ | $0.57_{\pm 0.096}$ | $19.98_{\pm 1.94}$ | $0.55_{\pm 0.091}$ | $27.53_{\pm 2.98}$ | $0.74_{\pm 0.085}$ |
| NoiseRgl | $21.61_{\pm 2.27}$ | $0.50_{\pm 0.073}$ | $18.09_{\pm 1.05}$ | $0.27_{\pm 0.036}$ | $28.04_{\pm 1.46}$ | $0.66_{\pm 0.037}$ |
| Orthogonal | $26.14_{\pm 5.79}$ | $0.78_{\pm 0.134}$ | $25.10_{\pm 5.08}$ | $0.77_{\pm 0.132}$ | $29.29_{\pm 5.32}$ | $0.84_{\pm 0.108}$ |
| G layers | $27.99_{\pm 5.70}$ | $0.83_{\pm 0.128}$ | $25.48_{\pm 4.76}$ | $0.78_{\pm 0.149}$ | $32.41_{\pm 4.60}$ | $0.90_{\pm 0.079}$ |

Table 2: Ablation study of the Gaussianization layers.

| Method | | | Accl = 8x, SNR = 20 dB | | Accl = 8x, SNR = 10 dB | |
| --- | --- | --- | --- | --- | --- | --- |
| | | | PSNR↑ | SSIM↑ | PSNR↑ | SSIM↑ |
| ICA (✗), | YJ (✗), | Lambt (✗) | $26.93_{\pm 6.40}$ | $0.787_{\pm 0.153}$ | $22.98_{\pm 5.54}$ | $0.604_{\pm 0.252}$ |
| ICA (✗), | YJ (✓), | Lambt (✗) | $26.92_{\pm 6.42}$ | $0.787_{\pm 0.156}$ | $23.14_{\pm 5.44}$ | $0.622_{\pm 0.245}$ |
| ICA (✗), | YJ (✗), | Lambt (✓) | $25.33_{\pm 5.89}$ | $0.743_{\pm 0.163}$ | $23.58_{\pm 5.19}$ | $0.695_{\pm 0.189}$ |
| ZCA, | YJ (✗), | Lambt (✗) | $27.08_{\pm 6.52}$ | $0.786_{\pm 0.157}$ | $22.94_{\pm 5.50}$ | $0.623_{\pm 0.235}$ |
| ICA (✓), | YJ (✗), | Lambt (✗) | $27.91_{\pm 5.77}$ | $0.824_{\pm 0.129}$ | $25.22_{\pm 4.76}$ | $0.770_{\pm 0.154}$ |
| ICA (✓), | YJ (✓), | Lambt (✗) | $27.99_{\pm 5.70}$ | $0.831_{\pm 0.128}$ | $25.48_{\pm 4.76}$ | $0.779_{\pm 0.149}$ |
| ICA (✓), | YJ (✗), | Lambt (✓) | $27.21_{\pm 5.74}$ | $0.816_{\pm 0.125}$ | $25.57_{\pm 4.98}$ | $0.779_{\pm 0.148}$ |
| ICA (✓), | YJ (✓), | Lambt (✓) | $27.37_{\pm 5.90}$ | $0.819_{\pm 0.135}$ | $25.09_{\pm 4.91}$ | $0.771_{\pm 0.144}$ |

## 4.2 IMAGE DEBLURRING USING GLOW

The mathematical model for deblurring is

$$\mathbf{d} = \boldsymbol{H} * \mathbf{m} + \boldsymbol{\epsilon}, \tag{11}$$

where $\boldsymbol{H}$ is a smoothing filter and $*$ denotes convolution. We used a Gaussian smoothing filter with a standard deviation of 3, and added noise $\boldsymbol{\epsilon} \sim \mathcal{N}(\mathbf{0}, 50^2 \mathbf{I})$ to the observed data. Although the system may not be under-determined, the high-frequency information is lost due to low-pass filtering; hence this is also an ill-posed problem. Though we only tested on facial images, deblurring has wide applications in astronomy and geophysics. See App. A.2 for more background information.

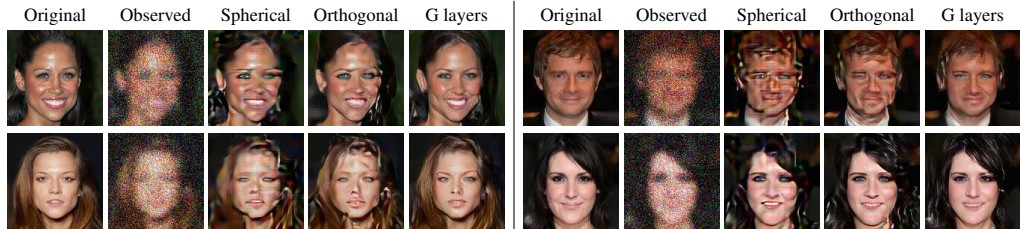

Figure 5: Comparison of deblurring results from different methods.

Table 3 and Fig. 5 show that the Gaussianization layers are also effective in Glow, better than using the spherical constraint or the orthogonal reparameterization. We also demonstrate the efficacy of Gaussianization layers when the forward model is inaccurate, in which case the induced error in data is not Gaussian (App. C).

**Ablation study** Table 3 also shows that the two parameterization schemes (App. D) for Glow using the Gaussianization layers have similar performance. Besides, we report the ablation study on the components of the Gaussianizaiton layers for Glow (App. C).

### 4.3 Eikonal tomography using StyleGAN2

In acoustic wave imaging (e.g., ultrasound tomography), we excite waves using sparsely distributed sources one at a time at the boundary of the object. Then we reconstruct its internal structures (the spatial distribution of wave speed) given the first-arrival travel time recorded on the boundary. The following eikonal equation approximately describes the shortest travel time $T(\boldsymbol{x}; \boldsymbol{x}_s)$ that the acoustic wave emerging from the source location $\boldsymbol{x}_s$ takes to reach location $\boldsymbol{x}$ inside the target object (Yilmaz, 2001):

$$|\nabla T(\boldsymbol{x}; \boldsymbol{x}_s)| = 1/c(\boldsymbol{x}), \quad T(\boldsymbol{x}_s; \boldsymbol{x}_s) = 0, \tag{12}$$

where $c(\boldsymbol{x})$ is the wave propagation speed at each location. Both this eikonal PDE and the implicitly defined forward mapping $c(\boldsymbol{x}) \rightarrow T(\boldsymbol{x})$ are nonlinear, and there has been little research on DGM-regularized inverse problems with such nonlinear characteristics. The inverse problem is severely ill-posed, which is equivalent to a curved-ray tomography problem. See App. A.3 for more background information. We added noise to the recorded travel time using the following formula: $T_{\text{noisy}}(\boldsymbol{x}_r; \boldsymbol{x}_s) = T(\boldsymbol{x}_r; \boldsymbol{x}_s)(1 + \epsilon)$, where $\epsilon \sim \mathcal{N}(0, 0.001^2)$ and $\boldsymbol{x}_r$ denotes any receiver location. In other words, longer traveltime corresponds to larger uncertainties.

We show that the Gaussianization layers outperformed other methods in this tomography task in Table 4 and Fig. 6. Note that this is a statistical conclusion. We also report an example where the spherical constraint works better than the Gaussianization layers (bottom right).

Table 3: Deblurring results using Glow.

| Method | LPIPS↓ | PSNR↑ | SSIM↑ |
|---|---|---|---|
| Spherical | $0.17_{\pm 0.06}$ | $21.78_{\pm 1.14}$ | $0.580_{\pm 0.066}$ |
| Orthogonal | $0.16_{\pm 0.06}$ | $21.91_{\pm 1.31}$ | $0.583_{\pm 0.063}$ |
| G layers P1 | $0.13_{\pm 0.05}$ | $22.40_{\pm 1.34}$ | $0.583_{\pm 0.069}$ |
| G layers P2 | $0.13_{\pm 0.05}$ | $22.47_{\pm 1.27}$ | $0.590_{\pm 0.064}$ |

Table 4: Eikonal tomography using StyleGAN2

| Method | PSNR↑ | SSIM↑ |
|---|---|---|
| TV | $20.66_{\pm 1.21}$ | $0.543_{\pm 0.073}$ |
| Spherical | $22.19_{\pm 4.05}$ | $0.686_{\pm 0.144}$ |
| G layers | $24.80_{\pm 2.55}$ | $0.803_{\pm 0.093}$ |

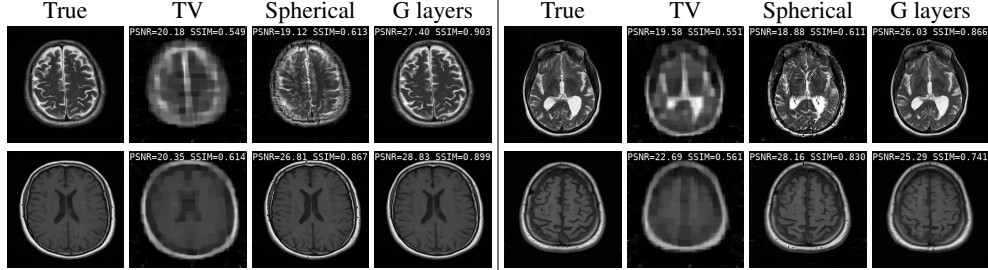

Figure 6: Comparison of eikonal tomography results from different methods.

## 5 Discussion and conclusions

We provide insights on the experiment results and further discuss this work's limitations, computational cost, and broader impact in App. K.

In summary, we have identified a critical problem in DGM-regularized inversion: the latent tensor can deviate from a typical example from the desired high-dimensional standard Gaussian distribution, leading to unsatisfactory inversion results. To address the problem, we have introduced the differentiable Gaussianization layers that reparameterize and Gaussianize latent tensors so that the inversion results remain plausible. In general, our method has achieved the best scores in terms of mean values and standard deviations compared with other methods, demonstrating our method's advantages and high performance in terms of accuracy and consistency. (Regarding the standard deviation of scores, we only compare with other methods with competitive mean scores, such as orthogonal reparameterization and spherical constraint.) Our proposed layers are plug-and-play, require minimal parameter tuning, and can be applied to various deep generative models and inverse problems.

## ACKNOWLEDGEMENTS

The author would like to thank Huseyin Denli, Ashutosh Tewari, Myun-Seok Cheon, Di Du, Stuart Harwood, Yu Fan, and Qiuzi Li for helpful discussions and the anonymous reviewers for their constructive feedback, which greatly improved the paper.

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

## A  BACKGROUND OF FORWARD MODELS

### A.1  COMPRESSIVE SENSING MRI

The MRI process essentially samples the spatial frequency components of a target, following some trajectories in the spatial frequency space (k-space) according to the design of the physical system. For example, a system may sample the k-space line-by-line horizontally/vertically or in radial directions. If the k-space has been fully sampled on a Cartesian grid, one can directly use inverse FFT to reconstruct the image. For various practical reasons, however, it is necessary to speed up the data collection process, usually by skipping data points in the k-space, which can be mathematically represented by a masking operation. In addition, there can be multiple coils with different sensitivity maps collecting data simultaneously. The mathematical formulation reads

$$\mathbf{d}_i = \boldsymbol{P}\boldsymbol{F}\boldsymbol{S}_i\mathbf{m} + \boldsymbol{\epsilon}, \quad i = 1, \cdots, N_{\text{coils}} \tag{13}$$

where $\mathbf{d}_i \in \mathbb{C}^N$ is the k-space data corresponding to the $i$-th coil, $\mathbf{m} \in \mathbb{R}^N$ is the target object, $\boldsymbol{P} \in \mathbb{R}^{N \times N}$ is the mask, $\boldsymbol{F} \in \mathbb{C}^{N \times N}$ is the Fourier transform operator, $\boldsymbol{S}_i \in \mathbb{C}^{N \times N}$ is the point-wise sensitivity map (a diagonal matrix) corresponding to the $i$-th coil, and $\boldsymbol{\epsilon}$ denotes noise.

To ensure a fair comparison with prior work and reproducibility, we used the same masks from the repository of Kelkar & Anastasio (2021) (Fig. 7). In addition, we also used the same single-coil setup as in Kelkar & Anastasio (2021), where the sensitivity matrix is an identity matrix.

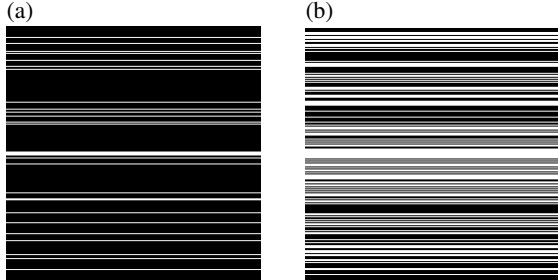

Figure 7: Masks for compressive sensing MRI (Kelkar & Anastasio, 2021). White: 1, black: 0. (a) $8\times$ acceleration; (b) $2\times$ acceleration.

We condense the effects of all operators into a linear operator $\boldsymbol{A} \in \mathbb{C}^{M \times N}$ and arrive at the under-determined system

$$\mathbf{d} = \boldsymbol{A}\mathbf{m} + \boldsymbol{\epsilon} \tag{10}$$

from the main text, where we use $\text{Accl} = N/M$ to denote the acceleration ratio.

### A.2  DEBLURRING

The mathematical model behind deblurring is

$$\mathbf{d} = \boldsymbol{H} * \mathbf{m} + \boldsymbol{\epsilon}, \tag{14}$$

where $\boldsymbol{H}$ is a smoothing filter, $*$ denotes convolution, and $\boldsymbol{\epsilon}$ is noise.

The purpose of deburring is to recover the original sharp image $\mathbf{m}$ given a noisy blurred observation $\mathbf{d}$. In this study, we showed deblurring examples for natural images. In scientific applications, deblurring or deconvolution is also a powerful tool. For example, in astronomy, $\mathbf{d}$ is a blurred image from a telescope, $\boldsymbol{H}$ is a point-spread function (PSF) constructed from the physics model of the telescope, and we want to obtain a sharper image from the observation (Starck et al., 2002). In geophysics, $\mathbf{d}$ can be the seismic data, $\boldsymbol{H}$ is a calibrated wavelet, and we want to obtain sharp images of reflectivities defining the boundaries of subsurface strata (Lines & Treitel, 1984; Zhang & Castagna, 2011). In general, $\boldsymbol{H}$ is a low-pass or band-pass filter, so certain frequency contents are lost in the forward process. The deblurring or deconvolution process needs to recover such missing information. In addition, the noise makes the inversion process unstable. The deblurring or deconvolution problem is thus ill-posed.

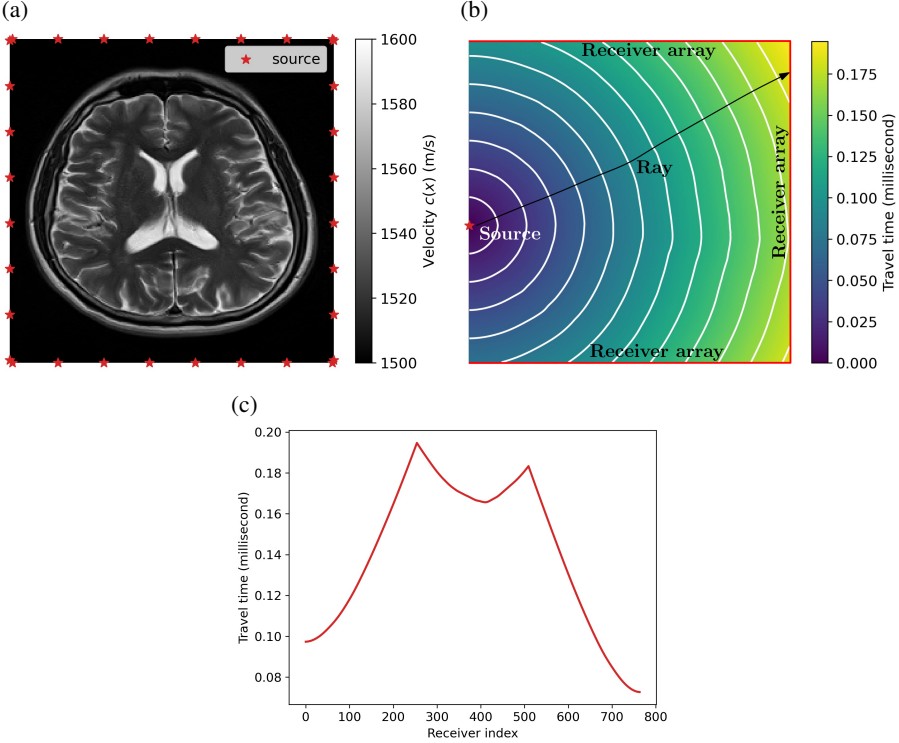

Figure 8: Experimental setup for eikonal tomography. (a) The target image and source locations; (b) Travel time field and the receiver array corresponding to the source. The contours are wavefronts. The propagation of waves can be viewed as curved rays traveling in directions perpendicular to such wavefronts; (c) The profile of the shortest wave travel time recorded at the receiver array. The receiver index starts from 0 in the top-left corner in subfigure (b) and increases in the clockwise direction.

### A.3 EIKONAL TOMOGRAPHY

In acoustic wave imaging (e.g., ultrasound tomography), we excite waves using sparsely distributed sources one at a time at the boundary of the object. Then we reconstruct its internal structures (the spatial distribution of wave speed) given the first-arrival travel time recorded on the boundary. The following eikonal equation describes the shortest travel time $T(\boldsymbol{x}; \boldsymbol{x}_s)$ that the acoustic wave emerging from the source location $\boldsymbol{x}_s$ takes to reach location $\boldsymbol{x}$ inside the target object in the high-frequency limit (Yilmaz, 2001):

$$|\nabla T(\boldsymbol{x}; \boldsymbol{x}_s)| = 1/c(\boldsymbol{x}), \quad T(\boldsymbol{x}_s; \boldsymbol{x}_s) = 0, \tag{15}$$

where $c(\boldsymbol{x})$ is the wave propagation speed at each location. Both this eikonal PDE and the implicitly defined forward mapping $c(\boldsymbol{x}) \rightarrow T(\boldsymbol{x})$ are nonlinear.

To be more specific, we show the setup of our experiment in Fig. 8(a). For the convenience of numerical testing, we put sources and receivers on the boundaries of a square box that contains the object, suggesting that the object is immersed in a square box filled with water. In reality, one can put sources and receivers directly on the target. The dimension of the square area is 25.6 cm × 25.6 cm with a grid interval of 0.001 m. There are eight sources located on each side, and receivers are located at each grid point on the boundary. Fig. 8(b) shows the shortest travel time field of the generated wave from the indicated source location. For each source, we only use receivers on the three other sides, indicated by the red lines, excluding the one on which the source is located. Fig. 8(c) shows the profile of the shortest wave travel time recorded at the receiver array. The receiver index starts from 0 in the top-left corner and increases in the clockwise direction.

We can interpret the nonlinearity and ill-posedness of eikonal tomography from another perspective. As shown in Fig. 8(b), we plot contours $T(\boldsymbol{x}) = \texttt{const}$ representing the wavefronts. Under the high-frequency approximation, the propagation of waves can be viewed as curved rays traveling in

directions perpendicular to such wavefronts. Since the wavefronts depend on velocity field $c(\boldsymbol{x})$, the rays are also functionals of the parameter $c(\boldsymbol{x})$ to be estimated, contrary to straight-ray tomography such as CT. The eikonal inversion problem is thus nonlinear. Besides, the rays carry information about the medium averaged along their paths. One property of curved-ray tomography is that the ray coverage is uneven inside the object. In fact, curved rays tend to avoid low-velocity areas, giving us little information about such regions, making the inverse problem intrinsically ill-posed.

We solved the eikonal equation using the fast sweeping method (Zhao, 2005) and computed the gradient using the discrete adjoint-state method, using the code from the repository of Li et al. (2020).

Our eikonoal tomography uses the same StyleGAN2 network as the compressive sensing MRI experiments. The value range of StyleGAN2 is $[-1, 1]$. In the forward model, we map the StyleGAN2 output to $c(\boldsymbol{x})$ in two steps. First, we convert its values to the range of $[0, 1]$ by using $\mathbf{m} \leftarrow (\mathbf{m}+1)/2$. Second, we convert the values to the range of acoustic wave velocity using $\mathbf{m} \leftarrow 100 \times \mathbf{m} + 1500$. This relationship is purely manufactured for our synthetic tests. One should use a more realistic relationship in practice.

## B  ADDITIONAL MOTIVATING EXAMPLES

**Glow**   We varied the $\ell_2$-norm of the latent tensors for Glow and reported the outputs in Fig. 9. The images are getting increasingly unrealistic as we increase the norm from $1.0\sqrt{d}$, where $d$ is the latent tensor dimension. This phenomenon demonstrates why we need to use a temperature $< 1$ for Glow.

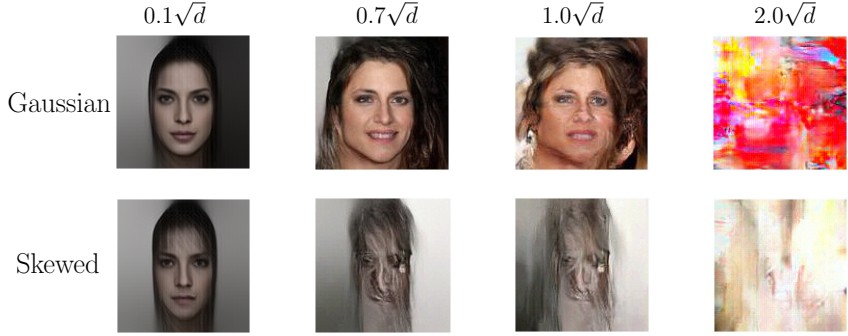

Figure 9: The visual effects of the $\ell_2$ norm of latent tensors on Glow outputs. The top and bottom panels correspond to latent tensors drawn from a spherical Gaussian and a spherical skewed distribution, respectively. The two tensors are scaled to have various norms: 0.1, 0.7, 1.0, and 2.0 of $\sqrt{d}$ (the square root of the latent tensor dimension). The images are getting increasingly unrealistic as we increase the norm from $1.0\sqrt{d}$. This phenomenon demonstrates why we need to use a temperature $< 1$ for Glow.

**StyleGAN2**   StyleGAN2 is a well-designed generator with a built-in spherical transformation for the style vector. It seems very robust to random vectors drawn from the distributions similar to those in Fig. 1. However, we were able to find challenging examples for it from inversion tasks. In Fig. 10, The first example is the direct output only using the pathologic style vector. The second example is the generator output using the style vector after whitening only. As we can see, whitening alone is ineffective in improving the image quality, which is confirmed by our ablation studies. On the contrary, if we use the ICA layer, we see a huge improvement in visual quality, and our interpretation is that getting rid of higher-order dependencies is very important. Note that there are eyeglasses in the third figure, meaning that if we relax the 1D Gaussian requirement, we can sample some rare examples. Finally, we use the full G layers to get another plausible image.

**Stable diffusion**   Stable Diffusion (Rombach et al., 2022) is a state-of-the-art deep generative model with text-to-image synthesis capability, which maps a Gaussian latent tensor to a high-resolution image.

No processing     Whitening only     ICA only     Full G layers

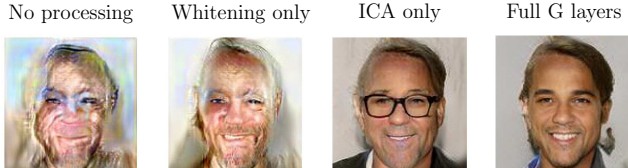

Figure 10: The visual effects of various components of Gaussianization layers on a pathologic style vector for StyleGAN2.

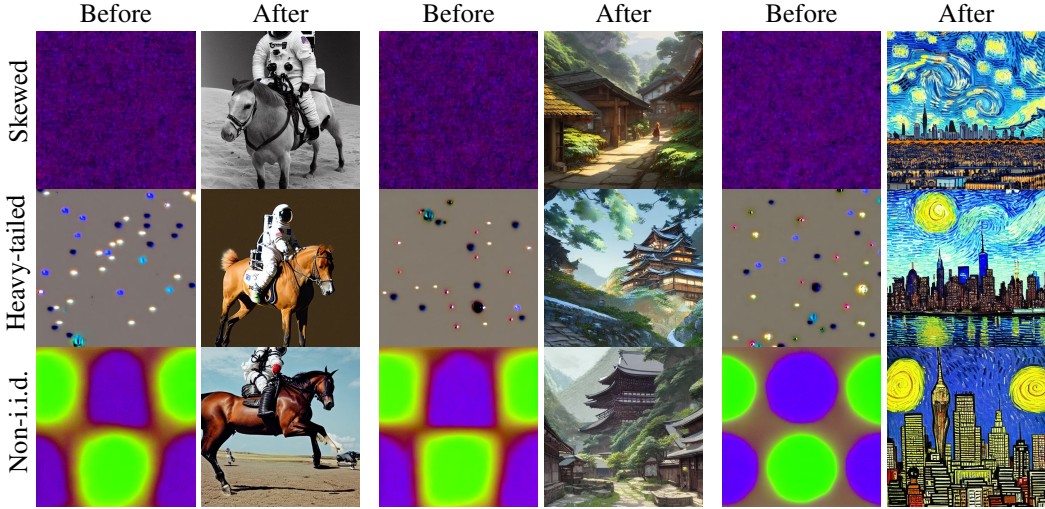

Figure 11: Comparisons of Stable Diffusion outputs before and after applying the Gaussianization layers to latent tensors. The components of these latent tensors are i.i.d. skewed, i.i.d. heavy-tailed, and non-i.i.d., respectively. We scaled all latent tensors to be on the sphere with a radius of $\sqrt{\text{tensor dimension}}$. Stable Diffusion picks up and amplifies the sparse large-amplitude points in the heavy-tailed case and the 2D sinusoidal pattern in the non-i.i.d. case.

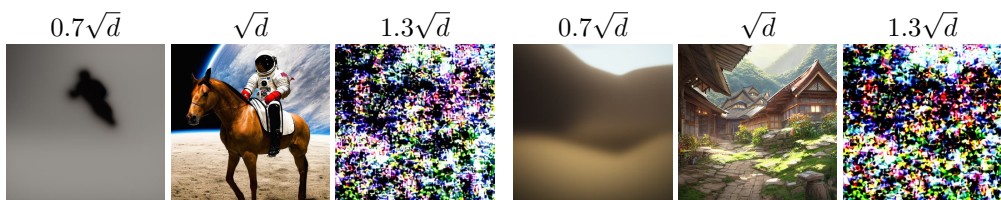

Figure 12: Comparison of Stable Diffusion outputs generated using spherical Gaussian latent tensors of different norms, where $d = \sqrt{\text{tensor dimension}}$.

We repeated the tests for Glow and StyleGAN2 on Stable Diffusion and reported the results in Fig. 11. The distribution for skewed case was the exponential-gamma distribution $\log \Gamma(1, 1)$ (or the "log-gamma distribution" in scipy), the distribution for the heavy-tailed case was a Lambert $W \times F_X$ distribution with parameter $\delta = 0.5$ based on a standard Gaussian (Goerg, 2015). To generate the latent tensor with non-i.i.d. entries, we first sampled $\mathbf{z}_0 \in \mathbb{R}^{4 \times 64 \times 64} \sim \mathcal{N}(\mathbf{0}, \mathbf{I})$. Then we added $0.5 * \sin(2\pi/64\, x) \otimes \cos(2\pi/64\, y)$ to each channel of $\mathbf{z}_0$ for the latent tensor $\mathbf{z}$, where $\otimes$ stands for outer product. The number of denoising steps, classifier-free guidance, and the output dimensions for Stable Diffusion are 50, 7.5, and $512 \times 512$, respectively.

The outputs of Stable Diffusion are completely out-of-range if the latent tensors deviate from standard Gaussian, with much poorer quality than those from Glow and StyleGAN2. For example, Stable Diffusion picks up and amplifies the sparse large-amplitude points in the heavy-tailed case and the 2D sinusoidal pattern in the non-i.i.d. case. However, with our Gaussianization layers, we are still able to constrain the outputs within the plausible range. Interestingly, we show in Fig. 12 that Stable Diffusion is much more sensitive to the norm of latent tensors than the other DGMs. Therefore, we anticipate that our Gaussianization layers are critical for potential inversion applications using Stable Diffusion such as text-guided inversion.

## C  ADDITIONAL EXPERIMENTS

First, we report in Table 5 an ablation study on the effects of various combinations of regularization techniques on style vectors and noise maps in StyleGAN2. The observations are (1) If we turn off the update of noise maps, the Gaussianization layers lead to the best result; (2) If we turn on the update of noise maps, applying the Gaussianization layers to both the style vectors and noise maps gives the best results.

Table 5: Ablation study on different combinations of regularization on the style vectors and noise maps in StyleGAN2 for compressive sensing MRI. In the parentheses, "u" stands for unconstrained, "s" stands for spherical constraint, "g" stands for reparameterization with the Gaussianization layers, "o" stands for orthogonal reparameterization, and "✗" means that the update is turned off. Note that StyleGAN2 has a built-in spherical transformation layer for the style vectors.

| Method | Accl = 8x, SNR = 20 dB | | Accl = 8x, SNR = 10 dB | |
| --- | --- | --- | --- | --- |
| | PSNR↑ | SSIM↑ | PSNR↑ | SSIM↑ |
| Style (s), Noise (✗) | $26.58_{\pm 5.90}$ | $0.80_{\pm 0.137}$ | $25.57_{\pm 5.24}$ | $0.78_{\pm 0.137}$ |
| Style (g), Noise (✗) | $27.39_{\pm 5.26}$ | $0.83_{\pm 0.124}$ | $27.02_{\pm 4.57}$ | $0.82_{\pm 0.122}$ |
| Style (o), Noise (✗) | $25.88_{\pm 5.74}$ | $0.78_{\pm 0.134}$ | $25.39_{\pm 5.09}$ | $0.77_{\pm 0.131}$ |
| Style (s), Noise (u) | $26.51_{\pm 6.51}$ | $0.77_{\pm 0.163}$ | $22.49_{\pm 5.62}$ | $0.57_{\pm 0.266}$ |
| Style (g), Noise (o) | $27.43_{\pm 6.04}$ | $0.81_{\pm 0.134}$ | $24.93_{\pm 4.27}$ | $0.76_{\pm 0.128}$ |
| Style (o), Noise (o) | $26.14_{\pm 5.79}$ | $0.78_{\pm 0.134}$ | $25.10_{\pm 5.08}$ | $0.77_{\pm 0.132}$ |
| Style (o), Noise (g) | $26.04_{\pm 5.70}$ | $0.78_{\pm 0.134}$ | $25.22_{\pm 5.12}$ | $0.77_{\pm 0.133}$ |
| Style (g), Noise (g) | $27.99_{\pm 5.70}$ | $0.83_{\pm 0.128}$ | $25.48_{\pm 4.76}$ | $0.78_{\pm 0.149}$ |

Second, we studied the effect of patch size on the performance of Gaussianization layers. To be more specific, we tested the effects of various patch sizes on 1D style vectors. The largest patch size is 64 since the number of extracted patches should not be smaller than the dimension of the patches. We only turned on the ICA layer and the standardization layer in this experiment. One can observe that both the PSNR and the SSIM increase as the patch size increases. We would advise using the largest possible patch size in Gaussianization layers, although we only used a patch size of 32 for style vectors in all other experiments.

In all experiments, we fixed the patch size for Glow as $3 \times 8 \times 8$ and the patch size for the noise maps in StyleGAN2 as $1 \times 8 \times 8$. In the experiments with Glow, the image dimension was $3 \times 128 \times 128$. If we chose a larger patch size, we could not obtain enough patch vectors. As for StyleGAN2, if we look at the parameterization illustrated in Fig. 14, there is already a $4 \times 4$ noise map unconstrained if we choose the patch size as $8 \times 8$. If we increase the patch size to $1 \times 16 \times 16$, there will be two

additional $8 \times 8$ noise maps unconstrained. To minimize the influence of unconstrained parameters, we chose the patch size for noise maps as $1 \times 8 \times 8$.

Table 6: Ablation study on patch size of the style vectors in StyleGAN2.

| Method | PSNR↑ | SSIM↑ |
|---|---|---|
| Patch size = 8 | $25.33_{\pm 6.42}$ | $0.76_{\pm 0.155}$ |
| Patch size = 16 | $26.83_{\pm 6.57}$ | $0.79_{\pm 0.144}$ |
| Patch size = 32 | $27.91_{\pm 5.77}$ | $0.82_{\pm 0.129}$ |
| Patch size = 64 | $28.03_{\pm 5.44}$ | $0.83_{\pm 0.130}$ |

Third, we did a parameter sweep on the weighting parameter $\beta$ in the Glow-regularized deblurring problem (formulation 3 or 44). Consistent with Fig. 20, the conventional Glow-based regularization is not as effective as our Gaussianization layers. If $\beta$ is too large (*e.g.*, 10 or 100), the inversion results become unrealistic with huge errors.

Table 7: Glow-regularized deblurring results using different $\beta$s.

| Method | LPIPS↓ | PSNR↑ | SSIM↑ |
|---|---|---|---|
| $\beta = 0.0$ | $0.172_{\pm 0.06}$ | $21.74_{\pm 1.27}$ | $0.58_{\pm 0.072}$ |
| $\beta = 0.01$ | $0.172_{\pm 0.06}$ | $21.75_{\pm 1.15}$ | $0.58_{\pm 0.067}$ |
| $\beta = 0.1$ | $0.174_{\pm 0.06}$ | $21.72_{\pm 1.15}$ | $0.57_{\pm 0.069}$ |
| $\beta = 1.0$ | $0.174_{\pm 0.06}$ | $21.95_{\pm 1.21}$ | $0.59_{\pm 0.067}$ |
| $\beta = 10.0$ | $0.247_{\pm 0.08}$ | $22.55_{\pm 1.10}$ | $0.58_{\pm 0.086}$ |
| $\beta = 100.0$ | $0.602_{\pm 0.15}$ | $12.56_{\pm 2.63}$ | $0.12_{\pm 0.101}$ |

Fourth, we investigated the performance of Gaussianization layers when the forward model is inaccurate (Table 8). Our Gaussianization layers still outperform the conventional Glow-regularized inversion.

Table 8: Glow-regularized deblurring with an inaccurate filter. The ground-truth standard deviation of the Gaussian filter is 3, which is used to generate the observed data, but we used 5 for inversion.

| Method | LPIPS↓ | PSNR↑ | SSIM↑ |
|---|---|---|---|
| Conventional $(\beta = 1.0)$ | $0.21_{\pm 0.06}$ | $17.72_{\pm 1.41}$ | $0.49_{\pm 0.059}$ |
| G layers | $0.17_{\pm 0.06}$ | $18.70_{\pm 1.40}$ | $0.50_{\pm 0.060}$ |

In addition, we did an ablation study on the components of Gaussianization layers in Glow-regularized deblurring (Table 9). We adopted the second parameterization scheme (App. D). The observations are similar to those from Table 2: 1. The ICA layer is the most significant part in improving result scores, especially LPIPS – the score that matches human perception the best; 2. The whitening/ZCA layer is not effective when used alone; 3. There is no clear winner among combinations of 1D Gaussianization layers, and their difference in performance is marginal. So we picked the Lambert $W \times F_X$ layer, one of the best-performing options, in all other deblurring experiments as the 1D Gaussianization layer.

**Inversion using out-of-distribution images** Finally, we tested the limit of DGM-regularized inversion by using real-world out-of-distribution target images. We randomly sampled 25 MRI images from the fastMRI DICOM dataset (Zbontar et al., 2018; Knoll et al., 2020). The RSNA clinical trial processor was used to anonymize the whole dataset, and each image has been manually inspected to make sure that there is no protected information. According to the fastMRI paper (Zbontar et al., 2018), "This dataset represents a larger variety of machines and settings than are present in the raw data." Also, the image values are represented in `uint16` rather than float numbers, and their value range can be very large (0 to several thousand). So we normalized the image values to be

Table 9: Ablation study of the components of Gaussianization layers applied to Glow-regularized deblurring.

| Method | | | LPIPS↓ | PSNR↑ | SSIM↑ |
|---|---|---|---|---|---|
| ICA (✗), | YJ (✗), | Lambt (✗) | $0.17_{\pm 0.06}$ | $21.79_{\pm 1.11}$ | $0.583_{\pm 0.065}$ |
| ICA (✗), | YJ (✗), | Lambt (✓) | $0.16_{\pm 0.06}$ | $21.96_{\pm 1.31}$ | $0.588_{\pm 0.065}$ |
| ICA (✗), | YJ (✓), | Lambt (✗) | $0.17_{\pm 0.06}$ | $21.75_{\pm 1.25}$ | $0.579_{\pm 0.067}$ |
| ICA (✗), | YJ (✓), | Lambt (✓) | $0.16_{\pm 0.06}$ | $21.99_{\pm 1.20}$ | $0.587_{\pm 0.066}$ |
| ZCA, | YJ (✗), | Lambt (✗) | $0.16_{\pm 0.05}$ | $21.66_{\pm 1.28}$ | $0.579_{\pm 0.070}$ |
| ICA (✓), | YJ (✗), | Lambt (✗) | $0.14_{\pm 0.06}$ | $22.52_{\pm 1.30}$ | $0.586_{\pm 0.070}$ |
| ICA (✓), | YJ (✗), | Lambt (✓) | $0.13_{\pm 0.05}$ | $22.47_{\pm 1.27}$ | $0.590_{\pm 0.064}$ |
| ICA (✓), | YJ (✓), | Lambt (✗) | $0.14_{\pm 0.05}$ | $22.49_{\pm 1.35}$ | $0.586_{\pm 0.070}$ |
| ICA (✓), | YJ (✓), | Lambt (✓) | $0.13_{\pm 0.06}$ | $22.40_{\pm 1.27}$ | $0.586_{\pm 0.066}$ |

between range $[-1, 1]$ using `img = (img - img.min())/(img.max()-img.min())*2 - 1`. However, the trained StyleGAN2 outputs are always slightly beyond $[-1, 1]$, and we clipped the values in both test image generation and inversion. Therefore, the new examples are out-of-distribution.

We kept the same experimental setup of $8\times$ acceleration and 20 dB SNR and tested inversion using the spherical constraint and Gaussianization layers on both only using the style vectors and using the additional noise maps. We report the inversion results in Table 10 and Fig. 13. To save experiment time, we only use one 1D Gaussianization layer (either Yeo-Johnson or Lambert $W \times F_X$).

Table 10: MRI inversion using real-world out-of-distribution images.

| Method | PSNR↑ | SSIM↑ |
|---|---|---|
| (a) Style only, spherical | $26.14_{\pm 3.59}$ | $0.784_{\pm 0.062}$ |
| (b) Style+noise, spherical | $25.26_{\pm 3.37}$ | $0.755_{\pm 0.065}$ |
| (c) Style only, G layers, YJ for 1D | $25.23_{\pm 3.30}$ | $0.765_{\pm 0.050}$ |
| (d) Style + noise, G layers, YJ for 1D | $25.39_{\pm 3.28}$ | $0.769_{\pm 0.045}$ |
| (e) Style only, G layers, Lambert for 1D | $25.39_{\pm 3.38}$ | $0.762_{\pm 0.064}$ |
| (f) Style + noise, G layers, Lambert for 1D | $25.45_{\pm 3.39}$ | $0.765_{\pm 0.064}$ |

First, none of the inversion results shown here are visually comparable to the ground truth, which shows the necessity of having a large training dataset that covers the target distribution. Also, it is vital to ensure that the pre-processing and value ranges of training and target images are the same. Second, in terms of metrics, the results are comparable to but slightly worse than those on synthetic test images reported previously. This is expected because these images are out of distribution. Besides, in cases where we applied Gaussianization layers, using both noise maps and style vectors gives better results than using style vectors only. In addition, Gaussianization layers improved results when we also optimized the noise maps in addition to the style vectors. However, we found that only using the style vector and the spherical constraint gives the best result on this test set, which contradicts our results reported in Table 5. Our interpretation is that the dimension of style vectors is still relatively lower than noise maps or the latent tensor used in Glow, so our Gaussianization layers may be more effective in dealing with the latter two types of parameters.

## D    REPARAMETERIZATION SCHEMES FOR STYLEGAN2 AND GLOW

Fig. 14 illustrates the reparameterization scheme for StylgeGAN2 using the Gaussianization layers. The patch size for the style vectors and noise maps are 32 and $1 \times 8 \times 8$, respectively. Note that we transpose the latent tensors in Fig. 14 compared to the notations in corresponding equations and algorithms. Also, as pointed out in Gu et al. (2020), we also find that a single latent code is insufficient for image reconstruction. Thus, we use multiple style vectors that were fed into different intermediate layers.

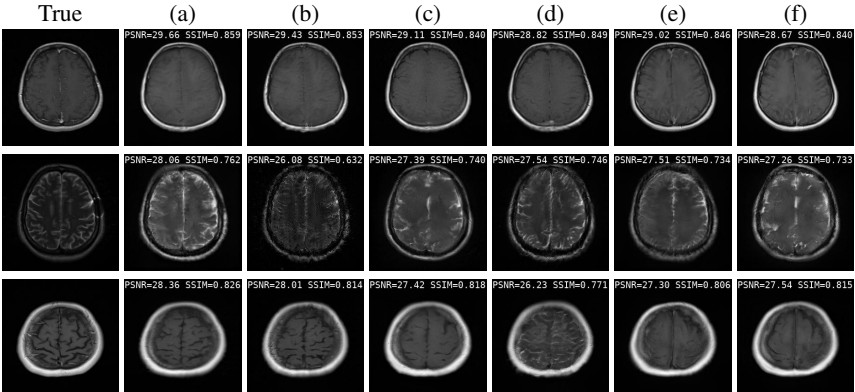

Figure 13: Examples of MRI inversion using out-of-distribution images. (Accl=8x, SNR=20 dB). (a) Style vector only + spherical constraint; (b) style vector + noise maps + spherical constraint; (c) style vector only + Gaussianization (only the Yeo-Johnson layer for 1D Gaussianization); (d) style vector + noise maps + Gaussianization (only the Yeo-Johnson layer for 1D); (e) style vector only + Gaussianization (only the Lambert layer for 1D); (f) style vector + noise maps + Gaussianization (only the Lambert layer for 1D).

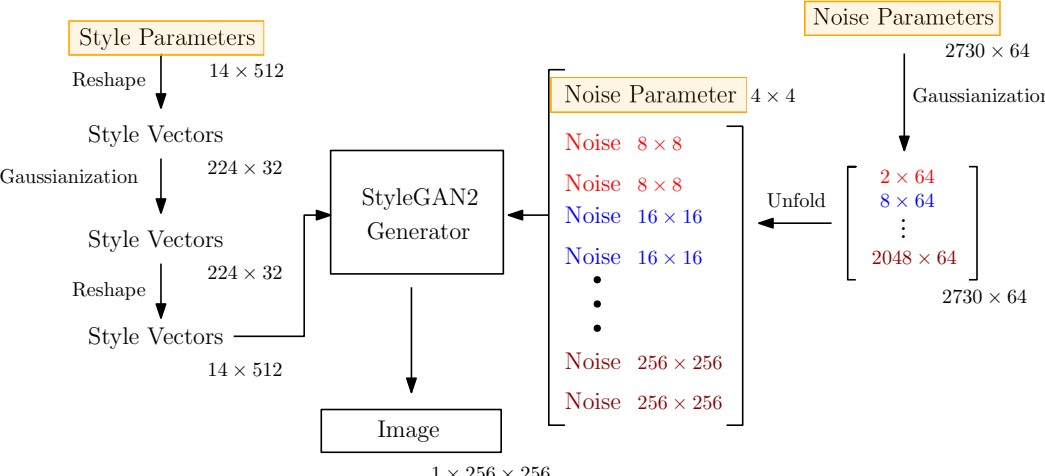

Figure 14: Reparameterization scheme for StyleGAN2. The dimensons of the latent parameters are [the number of vectors $\times$ vector dimension], except the $4 \times 4$ one. Note that we transpose latent tensors in corresponding equations and algorithms. The style and noise parameters before the Gaussianization layers are $\{\mathbf{v}_i|_{i=1,\cdots,N}\}$, and the style and noise vectors after the Gaussianization layers are $\{\mathbf{z}_i|_{i=1,\cdots,N}\}$ mentioned in the main text.

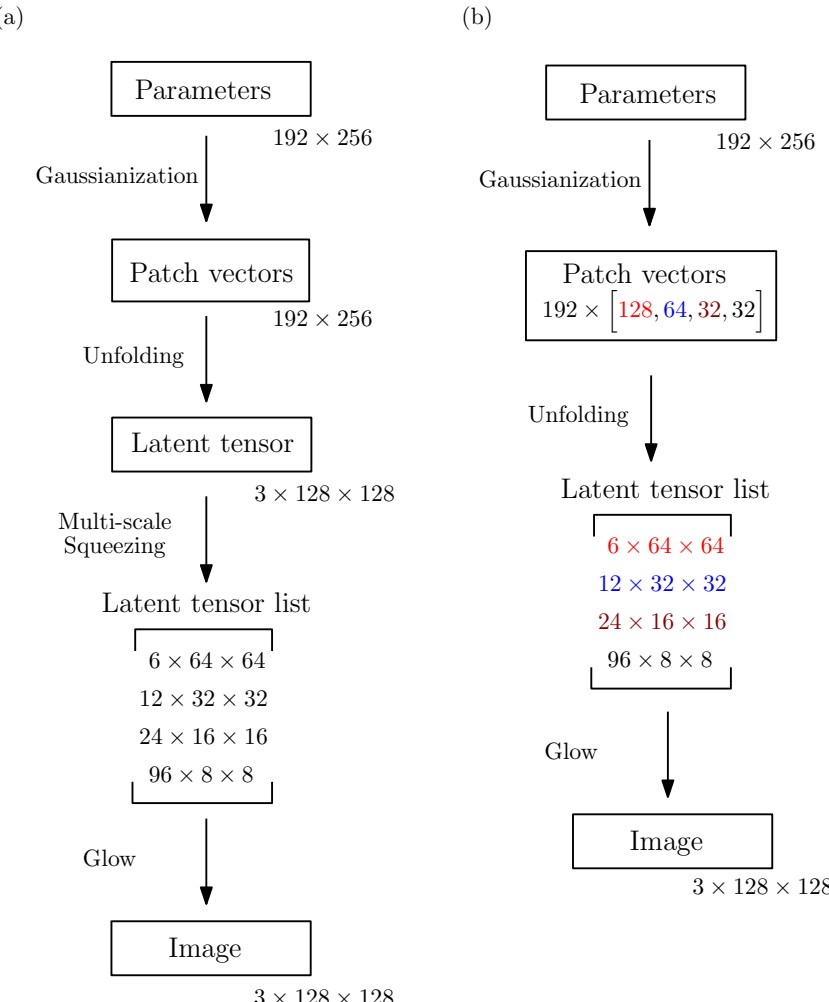

Figure 15: Two reparameterization schemes for Glow. (a) P1; (b) P2. The dimensions of the latent parameters are [vector dimension $\times$ the number of vectors]. The parameters before the Gaussianization layers are $\{\mathbf{v}_i|_{i=1,\cdots,N}\}$, and the patch vectors after the Gaussianization layers are $\{\mathbf{z}_i|_{i=1,\cdots,N}\}$ mentioned in the main text.

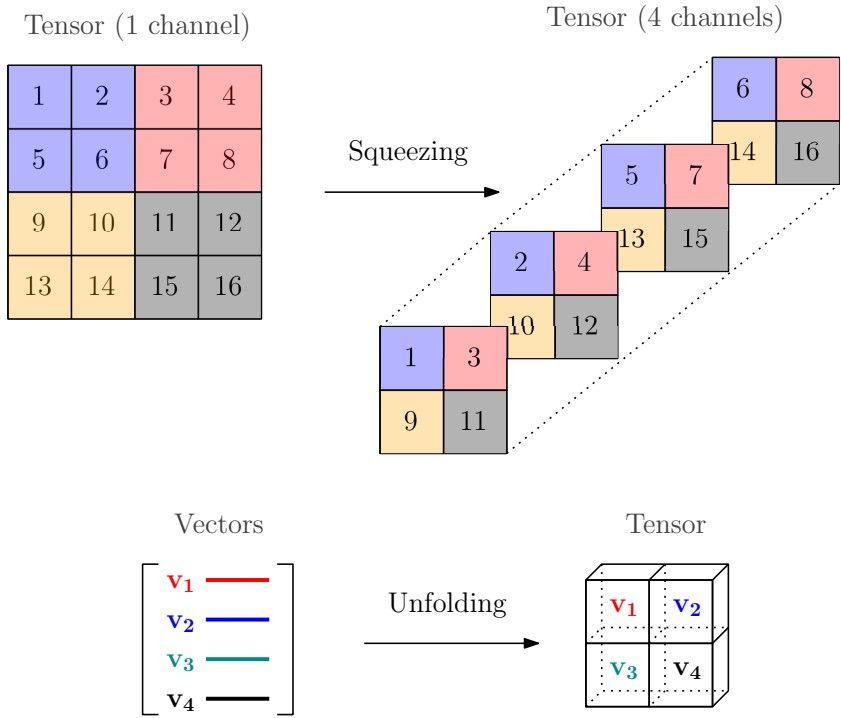

Figure 16: Illustration of the squeezeing and unfolding operations.

For Glow, we came up with two reparameterization schemes (Fig. 15). The first one (P1) reparameterizes patches from a latent tensor with the same dimension as the output image. The dimensions of the patches and the image are $3 \times 8 \times 8$ and $3 \times 128 \times 128$, respectively. Then a multi-scale squeezing operation maps the latent tensor into a list of tensors corresponding to different scales of Glow. Glow uses the list of tensors as the input. The second scheme (P2) reparameterizes patches extracted directly from tensors in the above list. We also partition each tensor into $3 \times 8 \times 8$ patches. The unfolding and squeezing operations are illustrated in Fig. 16. In the multi-scale architecture of Glow, the squeezing operation is applied recursively on half of the output tensors cut in the channel direction (Dinh et al., 2016; Kingma & Dhariwal, 2018).

# E  MORE DETAILS OF THE GAUSSIANIZATION LAYERS

## E.1  ICA LAYER

The overall ICA layer is summarized in algorithm 1. We set a maximum number of the fixed-point iterations to reduce computational cost and ensure accurate gradient computation that can pass the finite-difference convergence test. The gradients are backpropagated through the loops without using the trick introduced in App. G.

**Whitening**  The FastICA algorithm typically requires that the data are pre-whitened. Let $\boldsymbol{V}$ be the data matrix whose columns are data vectors. We first subtract the mean from the data vector using $\boldsymbol{V} \leftarrow \boldsymbol{V} - \texttt{mean}(\boldsymbol{V}, \texttt{dim} = 1)$. Then we compute the data covariance matrix using $\boldsymbol{C} = (1-\eta)\frac{1}{N-1}\boldsymbol{V}\boldsymbol{V}^\top + \eta\boldsymbol{I}$, where we use a small constant (e.g., $\eta = 0.001$) to blend the empirical covariance matrix and an identity matrix to avoid ill-conditioning.

After these data preparation steps, the ZCA whitening used throughout the study first computes the eigenvalues $\boldsymbol{\Lambda}$ and eigenvectors $\boldsymbol{D}$, and then output the whitened data using $\boldsymbol{V} \leftarrow \boldsymbol{D}\boldsymbol{\Lambda}^{-1/2}\boldsymbol{D}^\top\boldsymbol{V}$.

Alternatively, we can use the following steps to whiten the data, which are also used later in ICA iterations to decorrelate column vectors in the orthogonal matrix (Hyvarinen, 1999):

1. Initialize $\boldsymbol{W} \leftarrow \boldsymbol{I}$;
2. Compute

$$\boldsymbol{W} \leftarrow \boldsymbol{W}/\sqrt{\|\boldsymbol{W}^\top \boldsymbol{C} \boldsymbol{W}\|_2} \tag{16}$$

3. Repeat until convergence

$$\boldsymbol{W} \leftarrow \frac{3}{2}\boldsymbol{W} - \frac{1}{2}\boldsymbol{W}\boldsymbol{W}^\top \boldsymbol{C}\boldsymbol{W}, \tag{17}$$

4. Output $\boldsymbol{V} \leftarrow \boldsymbol{W}^\top \boldsymbol{V}$

---

**Algorithm 1:** ICA Layer

**Input** : Data matrix $\boldsymbol{V} \in \mathbb{R}^{D \times N}$; error tolerance $\epsilon$; damping parameters: $\eta = 10^{-4}$ and $\alpha = 0.8$; maximum iteration numbers $J = 10$ and $K = 100$.
**Output** : Matrix $\boldsymbol{P} \in \mathbb{R}^{D \times N}$ with i.i.d. entries
   // Whitening stage
1   $\boldsymbol{V} = \boldsymbol{V} - \mathtt{mean}(\boldsymbol{V}, \mathtt{dim} = 1), \boldsymbol{C} = (1 - \eta)\frac{1}{N-1}\boldsymbol{V}\boldsymbol{V}^\top + \eta\boldsymbol{I}$;
2   $\boldsymbol{V} \leftarrow \mathtt{ZCA\text{-}whitening}(\boldsymbol{V})$;
   // ICA stage
3   $\boldsymbol{W} = \boldsymbol{W}^* \leftarrow \boldsymbol{I}, \quad j \leftarrow 1$;
4   **while** $j \leq J$ **do**
5      $\boldsymbol{W} \leftarrow \frac{1}{N}\left[\alpha \boldsymbol{V}\phi\left(\boldsymbol{W}^\top \boldsymbol{V}\right)^\top - \boldsymbol{W}\mathtt{diag}\left(\phi'\left(\boldsymbol{W}^\top \boldsymbol{V}\right)\mathbf{1}\right)\right]$ Eq. 8 ;
6      $\boldsymbol{W} = \boldsymbol{W}_0 \leftarrow \boldsymbol{W}/\sqrt{\|\boldsymbol{W}^\top \boldsymbol{W}\|_2}, \quad k \leftarrow 1$;
7      **while** $k < K$ **do**
8         $\boldsymbol{W} \leftarrow \frac{3}{2}\boldsymbol{W} - \frac{1}{2}\boldsymbol{W}\boldsymbol{W}^\top \boldsymbol{W}$;
9         **if** $\|\boldsymbol{W} - \boldsymbol{W}_0\| < \epsilon$ **then**
10           break;
11         $\boldsymbol{W}_0 \leftarrow \boldsymbol{W}, \quad k \leftarrow k + 1$;
12      **if** $\|\boldsymbol{W} - \boldsymbol{W}^*\| < \epsilon$ **then**
13         break;
14      $\boldsymbol{W}^* \leftarrow \boldsymbol{W}, \quad j \leftarrow j + 1$;
15   **return** $\boldsymbol{P} \leftarrow \boldsymbol{W}^\top \boldsymbol{V}$.

---

**The modified FastICA iterations.** As stated in Hyvärinen (1999), the objective function for one neural unit of the weight vector $\boldsymbol{w}_i$ and input $\mathbf{v}$ is

$$\underset{\boldsymbol{w}_i}{\arg\max} \; \mathbb{E}\left[\Phi\left(\boldsymbol{w}_i^\top \mathbf{v}\right)\right], \text{ s.t., } \mathbb{E}\left[\left(\boldsymbol{w}_i^\top \mathbf{v}\right)^2\right] = 1, \tag{18}$$

where $\Phi$ is the contrast function (*e.g.*, logcosh). The original derivations convert this constrained optimization to an unconstrained one using Lagrange multipliers. However, this procedure is unnecessary since the matrix $\boldsymbol{W}$ is orthogonalized after each iteration, and the input vectors have been pre-whitened. Therefore, we only need to solve the following equation

$$\mathbb{E}\left[\mathbf{v}\phi\left(\boldsymbol{w}_i^\top \mathbf{v}\right)\right] = 0, \tag{19}$$

whose Jacobian is

$$\begin{aligned} J &= \mathbb{E}\left[\mathbf{v}\mathbf{v}^\top \phi'(\boldsymbol{w}_i^\top \mathbf{v})\right] \\ &\approx \mathbb{E}\left[\mathbf{v}\mathbf{v}^\top\right]\mathbb{E}\left[\phi'(\boldsymbol{w}_i^\top \mathbf{v})\right] = \mathbb{E}\left[\phi'(\boldsymbol{w}_i^\top \mathbf{v})\right], \end{aligned} \tag{20}$$

where $\phi$ is the derivative of $\Phi$. The Newton iteration scheme is thus

$$\boldsymbol{w}_i = \boldsymbol{w}_i - \mathbb{E}\left[\mathbf{v}\phi\left(\boldsymbol{w}_i^\top \mathbf{v}\right)\right]/\mathbb{E}\left[\phi'(\boldsymbol{w}_i^\top \mathbf{v})\right]. \tag{21}$$

To improve the convergence, we damp the iterations by a parameter $\alpha \in (0, 1)$. Also, using the same technique to convert the Newton iterations to fixed-point iterations in Hyvärinen (1999); Hyvarinen (1999), we arrive at the modified fixed-point iteration scheme:

$$\boldsymbol{w}_i = \alpha \mathbb{E}\left[\mathbf{v}\phi\left(\mathbf{w}_i^\top \mathbf{v}\right)\right] - \mathbb{E}\left[\phi'\left(\mathbf{w}_i^\top \mathbf{v}\right)\right]\boldsymbol{w}_i, \tag{22}$$

followed by the aforementioned decorrelation procedure after each step. The convergence of the modified FastICA iterations can be proved similarly as in Oja & Yuan (2006).

### E.2 Power transformation layer

---
**Algorithm 2:** Power Transformation Layer

**Input** : Data vector $\boldsymbol{p}$
**Output** : Vector $\boldsymbol{s}$ whose values are 1D Gaussianized.
1 Estimate $\lambda$ from $\boldsymbol{p}$ using Eq. 24 ;
2 Compute $\boldsymbol{s}$ with the estimated $\lambda$ and data $\boldsymbol{p}$ using Eq. 23 ;
3 **return** $\boldsymbol{s}$.

---

We propose to use the power transformation or Yeo-Johnson transformation (Yeo & Johnson, 2000) to reduce the skewness of distributions:

$$s(\lambda, p) = \begin{cases} \left((p+1)^\lambda - 1\right)/\lambda, & p \geq 0, \lambda \neq 0, \\ \log(p+1), & p \geq 0, \lambda = 0, \\ -([-p+1]^{2-\lambda} - 1)/[(2-\lambda)], & p < 0, \lambda \neq 2, \\ -\log(-p+1), & p < 0, \lambda = 2, \end{cases} \tag{23}$$

where $p$ is an input value, $s$ is an output value, and $\lambda$ is the parameter to be estimated. As shown in Fig. 3(a), the form of the Yeo-Johnson activation function depends on parameter $\lambda$. If $\lambda = 1$, the mapping is an identity mapping. If $\lambda > 1$, the activation function is convex, compressing the left tail and extending the right tail, reducing the left-skewness. If $\lambda < 1$, the activation function is concave, which oppositely reduces the right-skewness. The only parameter $\lambda$ is determined by solving an optimization problem that minimizes the negentropy:

$$\lambda = \arg\max_\lambda l(\lambda|\boldsymbol{p}) = \arg\max_\lambda -\frac{n}{2}\log\text{Var}\left(s\left(\lambda, p_i\right)\right) + (\lambda - 1)\sum_{i=1}^n \text{sign}(p_i)\log(|p_i| + 1), \tag{24}$$

where $\boldsymbol{p}$ is the input data vector with entries $p_{i,\,i=1,\cdots,n}$.

We use a custom operator based on SciPy's implementation using Brent's algorithm (Brent, 2013) to find an approximate minimum of Problem 24. Continuing from the approximate minimum, we use Brent's root finding algorithm (Brent, 2013) to find the minimum where the gradient vanishes.

Since the parameter $\lambda$ depends on input data, we need to back-propagate the gradient through the optimization process.

The power transformation layer is summarized in algorithm 2.

### E.3 Lambert $W \times F_X$ layer

Due to noise and inaccurate forward models, we observe that the distribution of latent vector values tends to be shaped as a heavy-tailed distribution during the inversion process. To reduce the heavy-tailedness, we adopt the Lambert $W \times F_X$ method detailed in Goerg (2015).

Let $X$ be a random variable whose CDF is $F_X$, with mean $\mu_X$ and standard deviation $\sigma_X$. The following transformation with a heavy-tail parameter $\delta \geq 0$:

$$S = \left(U\exp\left(\frac{\delta}{2}U^2\right)\right)\sigma_X + \mu_X, \tag{25}$$

where $U = (X - \mu_X)/\sigma_X$, is a bijection and maps $X$ to another random variable $S$ with heavier tails.

The transformation Eq. 25 is bijective if $\delta \geq 0$, and we can use the Lambert W function to find its inverse. The Lambert W function $W$ is defined as the inverse of $q = W^{-1}(t) = t \exp(t)$, where $t$ and $q$ are scalars. Given $q$, Halley's method can be used to find $t = W(q)$ (Corless et al., 1996). Hence, the inverse of Eq. 25 is

$$X = W_\delta \left( \frac{S - \mu_X}{\sigma_X} \right) \sigma_X + \mu_X, \tag{26}$$

where

$$W_\delta(u) = \text{sign}(u) \sqrt{\frac{W(\delta u^2)}{\delta}}. \tag{27}$$

We use the parameterized Lambert $W \times F_X$ distribution family to approximate a heavy-tailed input distribution and use Eq. 26 to recover a distribution with lighter tails. In order to make the recovered distribution close to a Gaussian distribution, we compute the optimal heavy-tail parameter $\delta$ by minimizing the difference between the kurtosis of the output distribution and 3 (Kurtosis is a common surrogate measure of negentropy (Hyvärinen & Oja, 2000)):

$$\hat{\delta}_{\text{GMM}} = \underset{\delta > 0}{\arg\min} \left| \text{Kurt} \left( W_\delta \left( \frac{\boldsymbol{s} - \mu_X}{\sigma_X} \right) \right) - 3 \right|^2, \tag{28}$$

where $\boldsymbol{s}$ is the data vector, and Kurt is the kurtosis. We constrain $\delta > 0$, and solve Eq. 28 using the L-BFGS-B optimizer (Zhu et al., 1997).

In addition, we estimate the mean $\mu_X$ and standard deviation $\sigma_X$ along with $\delta$ using the Iterative Generalized Method of Moments (IGMM) (Goerg, 2015), which embeds an optimization problem for $\delta$ in an outer loop of iterations to estimate $\sigma_X$ and $\mu_X$ (see algorithm 3). If the kurtosis of input data vector is not greater than 3, we skip the whole Lambert $W \times F_X$ layer by directly outputting the data vector.

---

**Algorithm 3:** Lambert $W \times F_X$ Layer with the Iterative Generalized Method of Moments (IGMM) (Goerg, 2015)

---

**Input** : Data vector $\boldsymbol{s}$, error tolerance $\epsilon = 10^{-5}$, maximum iteration number $K = 100$.
**Output** : vector $\boldsymbol{x}$ whose empirical distribution is less heavy-tailed than $\boldsymbol{s}$, and its kurtosis $\approx 3$)

1   Initialize: $\xi^{(0)} \leftarrow \left( \hat{\mu}_X^{(0)}, \hat{\sigma}_X^{(0)}, \hat{\delta}^{(0)} \right)$, $k = 0$;
2   Compute initial kurtosis $\beta_2 = \text{Kurt}(\boldsymbol{s})$ ;
3   **if** $\beta_2 \leq 3$ **then**
4     |   return $\boldsymbol{s}$;
5   **end**
6   **while** $k < K$ *and* $\|\xi^{(k)} - \xi^{(k-1)}\| \geq \epsilon$ **do**
7     |   $\boldsymbol{u}^{(k)} \leftarrow \left( \boldsymbol{s} - \mu_X^{(k)} \right) / \sigma_X^{(k)}$;
8     |   Compute $\delta^{(k+1)}$ using Eq. 28; $\boldsymbol{u}^{(k+1)} \leftarrow \boldsymbol{W}_{\delta^{(k+1)}} \left( \boldsymbol{u}^{(k)} \right)$; $\boldsymbol{x}^{(k+1)} \leftarrow \boldsymbol{u}^{(k+1)} \sigma_X^{(k)} + \mu_X^{(k)}$ ;
9     |   Update $\mu_X^{(k+1)} \leftarrow \mathbb{E} \left[ \boldsymbol{x}^{(k+1)} \right]$, and $\sigma_X^{(k+1)} \leftarrow \sqrt{\text{Var} \left( \boldsymbol{x}^{(k+1)} \right)}$ ;
10    |   $\xi^{(k+1)} \leftarrow \left( \hat{\mu}_X^{(k+1)}, \hat{\sigma}_X^{(k+1)}, \hat{\delta}^{(k+1)} \right)$ ;
11    |   $k \leftarrow k + 1$ ;
12   **end**
13   **return** $\boldsymbol{x} = W_\delta \left( \frac{\boldsymbol{s} - \mu_X}{\sigma_X} \right) \sigma_X + \mu_X,$

---

## F   Details of datasets and training

For MRI and eikonal tomography, we generate synthetic brain images as inversion targets using the pre-trained StyleGAN2 weights from Kelkar & Anastasio (2021), which are trained on data from the

databases of fastMRI (Zbontar et al., 2018; Knoll et al., 2020), TCIA-GBM (Scarpace et al., 2016), and OASIS-3 (LaMontagne et al., 2019). As a result, there is no sensitive personal information in our target images. In the data generation process, we only used one style vector of a dimension of 512. We picked 100 images that are visually plausible, whose random seeds can be found in our code. These random seeds were never used to initialize inversion. In inversion experiments, we used 14 such style parameters: one for the lowest resolution of $4 \times 4$, one for the tRGB layer, and two for each resolution from $8 \times 8$ to $256 \times 256$, to avoid inversion crime. This choice of expanding latent space dimension is also justified by our observation that only one style parameter vector is generally insufficient for inversion tasks with data from datasets such as CelebA-HQ (Karras et al., 2018).

We used the CelebA-HQ dataset (Karras et al., 2018) (under the Creative Commons CC BY-NC 4.0 license) for the deblurring experiments. All images were downsampled to the resolution of $128 \times 128$. We split the 30000 images from CelebA-HQ into the subsets of training (24183 images), validation (2993 images), and testing (2824 images) following the original splits from CelebA (Liu et al., 2015). For the inversion tests, we randomly selected 100 images from the test set.

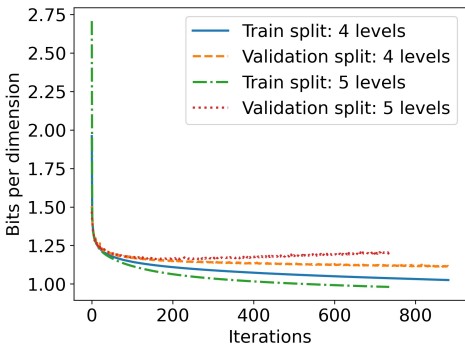

Figure 17: The negative log-likelihood or NLL (reported in bits per dimension) on the training and validation splits of the CelebA-HQ dataset with different numbers of multi-scale levels.

For the hyper-parameters of the Glow networks, we used 4 multi-scale levels and 32 flow-steps, and we only used additive coupling layers. Fig. 17 reports the training process. For each epoch, we computed the training negative log-likelihood (NLL) averaged throughout the epoch, and the validation NLL at the end of the epoch. The validation curves suggest that it is better to use 4 multi-scale levels. We chose the network weights from the epoch before the validation NLL stopped to decrease: 850 for the CelebA-HQ dataset. All training was conducted using $8 \times 32$ GB Nvidia V100 GPUs with a batch size of 64. We used the Adam optimizer (Kingma & Ba, 2015) with a learning rate of $10^{-4}$, as well as $\beta_1 = 0.9$ and $\beta_2 = 0.99$.

## G   GRADIENT COMPUTATION OF THE OPTIMIZATION-BASED DIFFERENTIABLE LAYERS

$$\arg\min_\theta l\left(h_\theta\left(\mathbf{x}\right)\right) \longrightarrow \theta^*$$

Input: $\mathbf{x}$ $\qquad h_{\theta^*}(\mathbf{x}) \longrightarrow$ Output: $\mathbf{y}$

Figure 18: Illustration of the forward computation of optimization-based ICA, Yeo-Johnson, and Lambert $W \times F_X$ layers. We denote the input and output by $\mathbf{x}$ and $\mathbf{y}$, respectively. The layer is represented by $h_\theta$, and the layer is defined by solving an optimization problem whose objective function is $l$. The back-propagation of gradients in components defined by Eq. 24 and Eq. 28 is enabled by the implicit function theorem and automatic differentiation detailed in App. G.

As shown in Fig. 18, in the power transformation and Lambert $W \times F_X$ layers, there are operators whose outputs are obtained by solving optimization problems formally described as

$$\boldsymbol{\theta}^* = \arg\min_{\boldsymbol{\theta}} l(\boldsymbol{x}, \boldsymbol{\theta}), \tag{29}$$

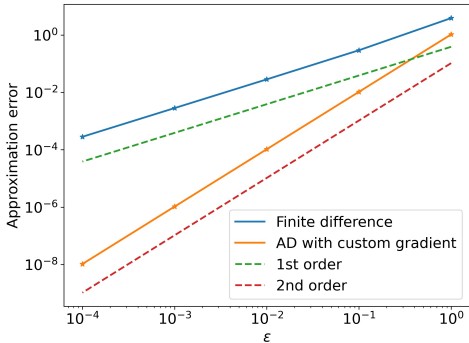

Figure 19: Gradient accuracy test for custom operators. An accurate gradient should make the error converge at a rate of the second order following the red dashed line.

where $l$ denotes the objective function that defines the operator (combining the objective function and layer $h_{\boldsymbol{\theta}}$ in Fig. 18), symbol $\boldsymbol{\theta}$ stands for the optimal output, a scalar in our cases but can also be a vector in general situations. The optimal condition is

$$l_{\boldsymbol{\theta}}(\boldsymbol{x}, \boldsymbol{\theta}) := L(\boldsymbol{x}, \boldsymbol{\theta}) = 0, \tag{30}$$

where the subscript denotes partial derivatives.

The optimal condition implicitly defines a forward operator of the following form:

$$\boldsymbol{\theta}^* = \mathrm{op}_{\mathrm{forward}}(\boldsymbol{x}). \tag{31}$$

The backward operator is

$$\frac{\partial \chi}{\partial \boldsymbol{x}} = \mathrm{op}_{\mathrm{backward}}\left(\frac{\partial \chi}{\partial \boldsymbol{\theta}}, \boldsymbol{\theta}^*, \boldsymbol{x}\right), \tag{32}$$

where $\chi$ is the objective function of an inverse problem.

Differentiating Eq. 30 with respect to $\boldsymbol{x}$, we have

$$L_{\boldsymbol{x}} + L_{\boldsymbol{\theta}}\boldsymbol{\theta}_{\boldsymbol{x}} = 0 \implies \boldsymbol{\theta}_{\boldsymbol{x}} = -L_{\boldsymbol{\theta}}^{-1}L_{\boldsymbol{x}}, \tag{33}$$

using the implicit function theorem. Then, to back-propagate the gradient from $\frac{\partial \chi}{\partial \boldsymbol{\theta}}$ to $\frac{\partial \chi}{\partial \boldsymbol{x}}$, we use

$$\frac{\partial \chi}{\partial \boldsymbol{x}} = \frac{\partial \chi}{\partial \boldsymbol{\theta}}\boldsymbol{\theta}_{\boldsymbol{x}} = -\frac{\partial \chi}{\partial \boldsymbol{\theta}}L_{\boldsymbol{\theta}}^{-1}L_{\boldsymbol{x}} = -\frac{\partial \chi}{\partial \boldsymbol{\theta}}H_{\boldsymbol{\theta}}^{-1}L_{\boldsymbol{x}}, \tag{34}$$

where $H_{\boldsymbol{\theta}}$ is the Hessian matrix of $\chi$ with respect to $\boldsymbol{\theta}$.

In our problems, the output $\boldsymbol{\theta}$ is a scalar, so it is easy to use automatic differentiation to compute $L_{\boldsymbol{\theta}}$ directly and hence $L_{\boldsymbol{\theta}}^{-1}$. Otherwise, if $\boldsymbol{\theta}$ has many parameters, we can first solve the following linear system with an auxiliary vector $\boldsymbol{\lambda}$:

$$\boldsymbol{\lambda}H_{\boldsymbol{\theta}} = -\frac{\partial \chi}{\partial \boldsymbol{\theta}}, \tag{35}$$

and then compute the gradient using

$$\frac{\partial \chi}{\partial \boldsymbol{x}} = \boldsymbol{\lambda}L_{\boldsymbol{x}}, \tag{36}$$

a technique also known as the adjoint-state method. Note that there is no need to compute the Hessian explicitly, but one can use automatic differentiation to compute the vector-Hessian product $\boldsymbol{\lambda}H_{\boldsymbol{\theta}}$ and utilize iterative linear solvers like GMRES (Saad & Schultz, 1986) to solve the linear system.

As a final note, we check the accuracy of our gradient computation using the finite-difference convergence test based on Taylor expansion:

$$f(\boldsymbol{x} + \epsilon \delta \boldsymbol{x}) = f(\boldsymbol{x}) + \epsilon \nabla f(\boldsymbol{x})^{\top}\delta \boldsymbol{x} + \mathcal{O}(\epsilon^2), \tag{37}$$

where $\delta \boldsymbol{x}$ is a random vector with a unit $\ell_2$ norm, and $\nabla f(\boldsymbol{x})$ is computed using our custom gradient. We here define $f$ as a composite function that maps the (vector) output of an forward operator to a scalar, e.g., $f(\boldsymbol{x}) = \|\mathrm{op}_{\mathrm{forward}}(\boldsymbol{x})\|_2^2$. Once we decrease $\epsilon$, we should see that the error term $f(\boldsymbol{x} + \epsilon \delta \boldsymbol{x}) - f(\boldsymbol{x}) - \epsilon \nabla f(\boldsymbol{x})^{\top}\delta \boldsymbol{x}$ decreases at a rate of the second order. All our layers passed this test, as the example shown in Fig. 19. This test should be conducted in double precision.

## H  ORTHOGONAL REPARAMETERIZATION

We also propose to reparameterize the latent vector $\mathbf{z}$ using an orthogonal matrix $\boldsymbol{R}$:

$$\mathbf{z} = \boldsymbol{R}\mathbf{v}, \ \mathbf{v} \sim \mathcal{N}(\mathbf{0}, \mathbf{I}), \tag{38}$$

where $\mathbf{v}$ is fixed during an inversion, and we treat $\boldsymbol{R}$ as the parameter instead. There are various ways to parameterized an orthogonal matrix. We choose the Cayley parameterization:

$$\boldsymbol{R} = (\boldsymbol{I} + \boldsymbol{W})(\boldsymbol{I} - \boldsymbol{W})^{-1}, \ \text{where} \ \boldsymbol{W} = -\boldsymbol{W}^{\top}, \tag{39}$$

one of the best reported in Liu et al. (2021). Therefore, the DGM-regularized inversion using orthogonal reparameterization is

$$\underset{\boldsymbol{W}}{\arg\min} \ (1/2) \|\mathbf{d} - f \circ g \, (\boldsymbol{R}\mathbf{v})\|_2^2. \tag{40}$$

The specific reparameterization schemes for StyleGAN2 and Glow are the same as in App. D. For style vectors, we use the full dimension of 512 because of the observation described in Table 6. For Glow and noise maps in StyleGAN2, we use patch sizes $3 \times 8 \times 8$ and $1 \times 8 \times 8$, respectively, to save computation time and memory.

## I  GAUSSIAN TYPICAL SET

The Gaussian annulus theorem states that a high-dimensional standard Gaussian distribution has most of its probability mass concentrated within an annulus area around a high-dimensional sphere:

**Theorem 1 (Gaussian Annulus Theorem (Blum et al., 2020))** *For an n-dimensional standard Gaussian, for any $\beta \leq \sqrt{n}$, all but at most $3e^{-c\beta^2}$ of the probability mass lies within the annulus $\sqrt{n} - \beta \leq |x| \leq \sqrt{n} + \beta$, where c is a fixed positive constant.*

This theorem gives a necessary geometric condition of typical samples from a high-dimensional standard Gaussian. However, the converse is not true: not all vectors whose $\ell_2$ norm is $\sqrt{n} - \beta \leq |x| \leq \sqrt{n} + \beta$ are typical examples from the standard Gaussian distribution. As a result, a DGM does not necessarily map a latent vector staying within the Gaussian annulus geometrically to a plausible image, which is demonstrated in Fig. 1.

The formal definition of a typical set is as follows.

**Definition 1 (Cover & Thomas (2012))** *Let $p_X(x)$ be a distribution whose support is $\mathcal{X}$. The typical set $A_\epsilon^{(n)}$ is defined as the set of sequences $(x_1, x_2, \cdots, x_n) \in \mathcal{X}^n$, $x_i \sim p_X$, that satisfy*

$$\left| H\left[X\right] + \frac{1}{n} \log p_X(x_1, \cdots, x_n) \right| \leq \epsilon, \tag{41}$$

*where $H\left[X\right]$ is the entropy of random variable X.*

Now a random vector $\mathbf{x} \in \mathbb{R}^n \sim \mathcal{N}(\mathbf{0}, \sigma^2\mathbf{I})$ can be factorized as i.i.d. random variables that are distributed as $\mathcal{N}(0, \sigma^2)$. Therefore, we can regard $\mathbf{x}$ as an i.i.d. sequence and give the following definition:

**Definition 2 (Gaussian Typical Set)** *A Gaussian typical set is the typical set $A_\epsilon^{(n)}$ of $\mathbf{x} \in \mathbb{R}^n \sim \mathcal{N}(\mathbf{0}, \sigma^2\mathbf{I})$.*

The following theorem guarantees that a typical sample from $\mathbf{x} \in \mathbb{R}^n \sim \mathcal{N}(\mathbf{0}, \sigma^2\mathbf{I})$ resides in the Gaussian typical set with very high probability.

**Theorem 2 (Cover & Thomas (2012))** *For every $\epsilon > 0$, the typical set has probability $P\left(A_\epsilon^{(n)}\right) > 1 - \epsilon$ with a sufficiently large dimension n.*

This theorem is a direct application of the asymptotic equipartition property (AEP), which is based on the i.i.d. assumption of sequence entries and the weak law of large numbers. Similar to the Gaussian annulus theorem, Theorem 2 depicts a geometric property of the Gaussian typical set: it is concentrated in an annulus near a shell of radius $\sigma\sqrt{n}$, which can be directly verified by definition. Of course, equivalently, if a vector whose $\ell_2$ norm significantly deviates from $\sqrt{n}$ cannot be a typical sample from a standard Gaussian. However, one cannot assert that a vector is sampled from the Gaussian typical set by only checking its norm.

## J  MISCELLANEOUS TOPICS

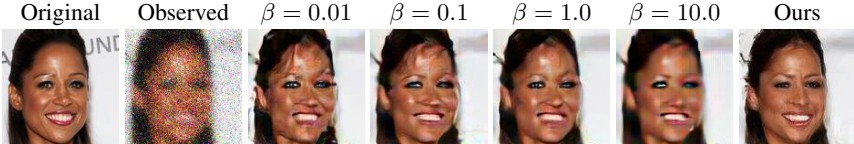

| Original | Observed | $\beta = 0.01$ | $\beta = 0.1$ | $\beta = 1.0$ | $\beta = 10.0$ | Ours |

Figure 20: Comparison of conventional DGM (Glow)-regularized deblurring results with different weighting factor $\beta$s and the result using our Gaussianization layers.

### J.1  INVERSION USING GLOW

In the Glow-regularization framework, we solve the inverse problem by finding the maximum *a posteriori* (MAP) estimate from

$$p_M(\mathbf{m}|\mathbf{d}) \propto p(\mathbf{d}|\mathbf{m})\, p_M(\mathbf{m}), \tag{42}$$

where $M$ denotes the parameter space. The probability density $p_M$ introduces our *a priori* knowledge and is represented by a normalizing flow (Glow) $g_{\boldsymbol{\theta}}$, which is a differentiable invertible mapping between two distributions, parameterized by neural network parameters $\boldsymbol{\theta}$: $\mathbf{m} = g_{\boldsymbol{\theta}}(\mathbf{z})$, where $\mathbf{z}$ is the latent vector. After training, the log probability density of a given model $\mathbf{m}$ is

$$\log p_M(\mathbf{m}; \boldsymbol{\theta}) = \log p_Z\left(g_{\boldsymbol{\theta}}^{-1}(\mathbf{m})\right) + \log\left|\det J_{g_{\boldsymbol{\theta}}^{-1}}(\mathbf{m})\right| = \log p_Z(\mathbf{z}) - \log\left|\det J_{g_{\boldsymbol{\theta}}}(\mathbf{z})\right|, \tag{43}$$

where $Z$ stands for the latent space. When we use the trained network in inversion, we freeze the network weights; hence, we drop $\boldsymbol{\theta}$ in $g_{\boldsymbol{\theta}}$ hereafter in our notation. Therefore, Eq. 3 for Glow-regularized inversion is

$$\arg\min_{\mathbf{z}}\ (1/2)\,\|\mathbf{d} - f \circ g(\mathbf{z})\|_2^2 - \beta\left(\log p_Z(\mathbf{z}) - \log\left|\det J_g(\mathbf{z})\right|\right). \tag{44}$$

The new regularization term in Eq. 3 is $\mathcal{R}'(\mathbf{z}) = -\beta\left(\log p_Z(\mathbf{z}) - \log\left|\det J_g(\mathbf{z})\right|\right)$, and we have shown the results with different $\beta$s in Fig. 20 and Table 7.

### J.2  DUALITY OF KL DIVERGENCE

As also shown in Papamakarios et al. (2017), the KL-divergence between two distributions does not change under a differentiable invertible transformation, so

$$D_{\mathrm{KL}}\left[p_M^*(\mathbf{m})\|p_M(\mathbf{m}; \boldsymbol{\theta})\right] = D_{\mathrm{KL}}\left[p_Z^*(\mathbf{z}; \boldsymbol{\theta})\|p_Z(\mathbf{z})\right], \tag{45}$$

where $p_M^*$ is the target distribution in the physical parameter space, and $p_Z^*$ is the corresponding latent-space distribution under the normalizing flow. This means that minimizing the forward KL divergence in the $M$ domain or physical parameter space is equivalent to minimizing the reverse KL-divergence in the $Z$ domain or the latent space.

This fact and Theorem 2 imply that a well-trained normalizing flow maps samples from the target distribution into the Gaussian typical set with very high probability and vice versa.

### J.3 INVARIANCE OF KL-DIVERGENCE AND MULTI-INFORMATION

For the completeness of this paper, we briefly prove the key properties of KL-divergence and multi-information used in the Gaussianization framework.

First, we prove that the KL-divergence is invariant under differentiable bijections. We write the definition of the KL-divergence between distribution $p_{\mathbf{x}}$ and $q_{\mathbf{x}}$ of random vector $\mathbf{x}$ as

$$D_{\mathrm{KL}}(p_{\mathbf{x}}\|q_{\mathbf{x}}) = \int p_{\mathbf{x}}(\boldsymbol{x}) \log \frac{p_{\mathbf{x}}(\boldsymbol{x})}{q_{\mathbf{x}}(\boldsymbol{x})} \mathrm{d}\boldsymbol{x}. \tag{46}$$

Suppose there is a differentiable and invertible transformation $T$ that maps $\mathbf{x}$ to $\mathbf{u}$: $\mathbf{u} = T(\mathbf{x})$. Then, we know the PDF under change of variable is

$$p_{\mathbf{x}}(\boldsymbol{x}) = p_{\mathbf{u}}(\boldsymbol{u})|\det J_T(\boldsymbol{x})|, \quad q_{\mathbf{x}}(\boldsymbol{x}) = q_{\mathbf{u}}(\boldsymbol{u})|\det J_T(\boldsymbol{x})|, \tag{47}$$

where we define the Jacobian matrix $J_T(\boldsymbol{x}) = \frac{\partial \mathbf{u}}{\partial \mathbf{x}}(\boldsymbol{x})$. Therefore, we have

$$
\begin{aligned}
D_{\mathrm{KL}}(p_{\mathbf{x}}\|q_{\mathbf{x}}) &= \int p_{\mathbf{x}}(\boldsymbol{x}) \log \frac{p_{\mathbf{x}}(\boldsymbol{x})}{q_{\mathbf{x}}(\boldsymbol{x})} \mathrm{d}\boldsymbol{x} \\
&= \int p_{\mathbf{u}}(\boldsymbol{u})|\det J_T(\boldsymbol{x})| \log \frac{p_{\mathbf{u}}(\boldsymbol{u})|\det J_T(\boldsymbol{x})|}{q_{\mathbf{u}}(\boldsymbol{u})|\det J_T(\boldsymbol{x})|} \mathrm{d}\boldsymbol{x} \\
&= D_{\mathrm{KL}}(p_{\mathbf{u}}\|q_{\mathbf{u}})
\end{aligned}
\tag{48}
$$

Then, we show that the multi-information does not change under component-wise invertible differentiable transformations. The multi-information is defined by the KL-divergence between the joint distribution $p(\mathbf{x})$ and the marginals:

$$I(\mathbf{x}) = I(\mathrm{x}_1, \cdots, \mathrm{x}_D) = D_{\mathrm{KL}}\left(p(\mathbf{x}) \,\Big\|\, \prod_j^D p_{\mathrm{x}}^j(\mathrm{x}_j)\right). \tag{49}$$

We define the bijection $\mathbf{u} = T(\mathbf{x})$ specifically with invertible differentiable transformations for each component $T_i(\mathrm{x}_i) = \mathrm{u}_i$. Then, using the KL-divergence invariance we just proved and that $\prod_j^D p_{\mathrm{x}}^j(\mathrm{x}_j) = \prod_j^D p_{\mathrm{u}}^j(\mathrm{u}_j)|\det J_T(\boldsymbol{x})|$, we get $I(\mathbf{x}) = I(\mathbf{u})$.

### J.4 THE ROLLING OPERATION

After applying one set of the Gaussianization layers to the latent tensor, it might be desirable to apply the layers to a different set of non-overlapping patches from the latent tensor again. The rolling operation shifts the latent tensor in the horizontal and vertical directions by half of the patch size $w$ before patch extractions. The values at the boundaries are wrapped around to the opposite sides. One can, for example, use the `torch.roll` command to implement this functionality.

## K ADDITIONAL DISCUSSIONS

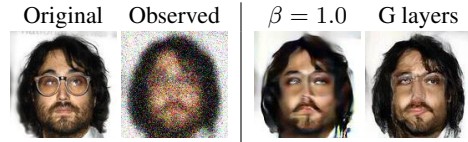

Original  Observed  $\beta = 1.0$  G layers

Figure 21: A failure example where inversion is unable to restore the eyeglasses because of the strong constraint from traversing within the Gaussian typical set. The results are dominated by the bias of the training dataset.

We point out the limitations and potential improvements of this work. First, requiring latent vectors to traverse within the Gaussian distribution/typical set means that the statistics of the training dataset will dominate the results. Fig. 21 shows that our method cannot restore the eyeglasses because

it is not a significant feature within our training data. One needs to pay particular attention to training dataset construction to ensure that it represents the typical features to be investigated. One should also interpret their results with this caveat in mind. On the other hand, one may combine Gaussianization layers and latent space manipulation techniques (Shen et al., 2020) for a guided inversion, *e.g.*, requiring that the inverted images should have eyeglasses. Second, we only use one set of Gaussianization layers. One can further improve the performance by using additional sets of layers, with possible different partition schemes, such as using the rolling operation (App. J.4) to shift the patch extraction locations. Third, since the deep generative models are highly nonlinear, the results may get stuck in local minima. We mitigate this problem by starting inversion from multiple randomly initialized latent vectors. However, this technique is computationally expensive. It is an important research direction to improve the initialization process.

The Gaussianization layers are based on optimization and fixed-point iterations, which can be computationally expensive. To study possible simplifications, we estimated the forward and backward time for various combinations of sub-layers using 100 repeated experiments (Table 11). The computation was performed using one Nvidia V100 GPU and an Intel Xeon Platinum 8276 CPU. We always turn on the ICA layer (together with whitening) since the ablation studies show that it contributes the most to performance increase (Table 2 and Table 9). Combining results from those ablation studies, we can see that the option ICA+YJ strikes a good balance between performance and computation time. It has a shorter runtime than ICA+Lambert and ICA+YJ+Lambert. At the same time, it gives comparable and sometimes the best results. On the other hand, the additional computation time from Gaussianization layers is well justified by the performance increase, especially if the physics simulation is much more time-consuming. For example, the eikonal solver for the experimental setup in our study has a forward runtime of $0.293\pm0.013$ seconds and a backward runtime of $5.933\pm0.151$ seconds, respectively. In addition, our implementation can also be greatly improved by more efficient utilization of GPUs and rewriting bottleneck components using high-performance codes.

Table 11: Forward and backward computation time of various combinations of Gaussianization sublayers. The times were computed using tensors of the same shape as latent tensors for Glow and StyleGAN2, respectively. Each computation time was estimated by 100 repeated experiments. We always turn on the whitening layer and the standardization layer.

| (Note: the time is measured in seconds) | ICA | ICA + YJ | ICA + Lambert | ICA + YJ + Lambert |
|---|---|---|---|---|
| Forward (Glow tensor) | $0.030_{\pm0.038}$ | $0.286_{\pm0.052}$ | $1.911_{\pm0.338}$ | $2.022_{\pm0.304}$ |
| Backward (Glow tensor) | $0.017_{\pm3.8e\text{-}4}$ | $0.022_{\pm3.3e\text{-}4}$ | $0.751_{\pm0.110}$ | $0.693_{\pm0.123}$ |
| Forward (StyleGAN2 tensor) | $0.033_{\pm3.1e\text{-}4}$ | $0.280_{\pm0.047}$ | $1.068_{\pm0.141}$ | $1.212_{\pm0.097}$ |
| Backward (StyleGAN2 tensor) | $0.027_{\pm2.6e\text{-}4}$ | $0.037_{\pm5.4e\text{-}4}$ | $1.077_{\pm0.100}$ | $0.964_{\pm0.065}$ |

We also provide insights into why Gaussianization layers perform better than other methods. First, the spherical constraint only relies on the geometric property of the Gaussian typical set (App. I). However, as shown in Fig. 1, it is insufficient to only constrain the norm of latent vectors in inversion. Gaussianization is necessary to ensure inverted results are in the range of DGMs. Second, the orthogonal reparameterization generally underperforms the Gaussianization layers. Our interpretation is as follows: rather than actively destroying latent tensor patterns like the Gaussianization layers do, orthogonal reparameterization cannot prevent permuting the fixed typical Gaussian latent tensors into ones with spatial patterns. In addition, we have shown in Fig. 10 that the whitening transformation alone is ineffective in keeping the DGM outputs within the range. We thus interpret that reducing higher-order dependencies is essential. Therefore, the noise regularization and the whitening layers yield less satisfying results than the Gaussianization layers.

We believe this study can benefit both the broader scientific community and the general public. However, we point out that the MRI and eikonal tomography experiments are purely synthetic and numerical. They do not fully reflect realistic medical imaging configurations. One should be cautious about applying our techniques to real data and interpreting the results.

