# OpenReview forum: "Differentiable Gaussianization Layers for Inverse Problems Regularized by Deep Generative Models"
_ICLR.cc/2023/Conference — ICLR 2023 poster_

### Official Review · Reviewer_k6hw · 2022-10-25

**Confidence:** 4
**Correctness:** 3
**Technical Novelty And Significance:** 3
**Empirical Novelty And Significance:** 3
**Recommendation:** 5

**Clarity, Quality, Novelty And Reproducibility:**

- It seems that MRI experiments are realistic or not. Many works ignore that MRI measurements are complex, not real, and this work also seems to make the same mistake. More realistic simulations for MRI including multi-coils (all modern MRI scanners are using multi-coils!) and complex Gaussian noise.
- Glow is an important work, but it seems like an outdated method as a compared method and there are a number of related recent works with better performance. Comparing with more recent methods will make this work stronger. For StyleGANv2, comparing with (Wulff & Torralba 2020) will help this work to be demonstrated as a SOTA method. Comparing with score matching based works will be surely a plus.
- Gaussianization technique is also used in other image generation works. See Yen-Chi Cheng et al., InOut : Diverse Image Outpainting via GAN Inversion, CVPR 2022. This work should survey various image generation works with Gaussianization not only for compressive sensing MRI, deblurring and eikonal tomography, but also for other related inverse problems.


**Strength And Weaknesses:**

Strength:
- Unless this work clearly resolved the first issue in the weakness, it is hard to write down any strength since it is unclear if those strengths are belonging to this work or other works.

Weakness:
- This work looks very similar to the following prior work: C Meng, Y Song, J Song, S Ermon, Gaussianization Flows, AIStats 2020. The key idea of this work is to reparameterize latent vectors using learnable orthogonal matrices (Cayley parameterization) while the work of Meng et al. also used "trainable orthogonal matrix layer." This is really critical in my assessment for this work and it is really unfortunate that this work missed this prior work. Thorough comparison with this prior work seems critical for this work to be justified as a new work.
- It is unclear if all comparisons in this work were proper. For example, (Wulff & Torralba 2020) also investigated on Gaussianization in the latent space for StyleGANv2 and this work should be compared with the proposed method with StyleGANv2 in terms of performance and the level of Gaussianization.
- It is unclear if the proposed methods worked as designed or not. All comparisons show that the proposed methods outperformed other selected methods, but there is no investigation on how much Gaussianization was achieved with the proposed methods. It will be helpful for many readers to see some concrete quantitative results on Gaussianization itself.


**Summary Of The Paper:**

This work proposes a method to reparameterize and gaussianize the latent tensors for deep generative models such as StyleGAN2 and Glow. This idea was evaluated in a number of inverse problems such as compressive-sensing MRI, image deblurring, and eikonal tomography.



**Summary Of The Review:**

This work introduced a Gaussianization layer for image generation models and then used it for solving a few inverse problems. This work was compared with the works not using Gaussianization. However, using Gaussianization for image generation is not new as discussed in the review and some of them look critical. Moreover, it is unclear if the intended Gaussianization really happened in all problems - no quantitative results for Gaussianization. Thus, in its current form, without justifying its novelty over (C Meng et al., AIStats 2020) and other works using the idea of Gaussianization, it will be very hard not to recommend rejection.

---

> ### Author Response · Authors · 2022-11-05
> **Response to Reviewer k6hw - 1**
>
> Thank you very much for your review and comments!
>
> **First**, we clearly state that our work differs from the prior work (C Meng et al., AIStats 2020).
> 1. The "learnable orthogonal matrices (Cayley parameterization)" that the reviewer found similar to the "trainable orthogonal matrix layer." in Meng et al. are **NOT** used in our Gaussianization layers. They are used in an alternative method -- orthogonal parameterization that was also proposed by us to compare with our main contribution -- the Gaussianization layers. This method is also different from Meng et al. Please see Appendix H for details. Confusing this alternative method with the Gaussianization layers would lead to some misunderstanding of the key idea of our paper.
> 2. Our proposed G layers are "implicit layers" similar to the OptNet [1] or the deep-equilibrium networks [2]. They are defined by solving optimization problems from ICA and 1D Gaussianizations. In contrast, Meng et al. is a type of normalizing flow with a totally different training procedure. In fact, their work does not use the two-step Gaussianization procedures explicitly as in our work but is inspired by such a principle to design the invertible layers for normalizing flows.
> 3. The details of building blocks are different. We use ICA to construct the orthogonal matrix, while Meng et al. explicitly state that they are not using ICA. Our 1D Gaussianization layers are based on power transformation and Lambert $W\times F_X$, while theirs is based on KDE parameterization and quantile transform.
> 4. Our method aims to Gaussianize latent tensors for better solving inverse problems, while the method of Meng et al. aims to model target distribution as a generative model. Our layers are not trained using a training dataset before inversion but are defined within each forward mapping in the inversion process.
>
>
> We actually cited Meng et al. in a previous version of this paper since we were inspired by their great work. We later removed it because we thought the Gaussianization principles are well covered in the pioneering works of Gaussianization [3] and RBIG [4] that Meng et al. is also based on. Now considering the reviewer's question, we will cite Meng et al. in the revised version.
>
> **In summary, the similarity between Meng et al. and our method is that we are both based on the Gaussianization principle proposed by [3] and [4]. However, our methods' purpose, essence, and architectures are different.**
>
> **Second**, we **did** compare our method with the other Gaussianization technique proposed by Wulff & Torralba 2020, which is also used in Yen-Chi Cheng et al. In fact, the CSGM-w method mentioned in [5] uses this Gaussianization technique. Our experiments show that CSGM-w does not yield satisfactory results. Similar poor results from this "Gaussianization" technique can be found in [5], and the authors had to introduce prior images for improvement. The technique from Wulff & Torralba 2020 is only a 1D Gaussianization (the empirical covariance matrix is estimated by fitting to the whole dataset, which may not be accurate for individual examples). In fact, a leaky Relu with a slope of 5.0 looks very similar to our power transformation layer with a $\lambda < 1$ that reduces right-skewness. However, as our motivating example and inversion experiments show, destroying the spatial pattern of latent tensors is also (more) crucial. In the revised version, we will add one more motivating example using **Stable Diffusion** to further demonstrate this point. Finally, the technique in Wulff & Torralba 2020 is empirical with no theoretical justifications, and it can only be applied to StyleGAN2 rather than deep generative models in general.
>
> [1] Amos, Brandon, and J. Zico Kolter. "Optnet: Differentiable optimization as a layer in neural networks." In International Conference on Machine Learning, pp. 136-145. PMLR, 2017.
>
> [2] Bai, Shaojie, J. Zico Kolter, and Vladlen Koltun. "Deep equilibrium models." Advances in Neural Information Processing Systems 32 (2019).
>
> [3] Chen, Scott, and Ramesh Gopinath. "Gaussianization." Advances in neural information processing systems 13 (2000).
>
> [4] Laparra, Valero, Gustavo Camps-Valls, and Jesús Malo. "Iterative gaussianization: from ICA to random rotations." IEEE transactions on neural networks 22, no. 4 (2011): 537-549.
>
> [5] Kelkar, Varun A., and Mark Anastasio. "Prior image-constrained reconstruction using style-based generative models." In International Conference on Machine Learning, pp. 5367-5377. PMLR, 2021.

---

> > ### Author Response · Authors · 2022-11-05
> > **Response to Reviewer k6hw - 2**
> >
> > > **1. "there is no investigation on how much Gaussianization was achieved with the proposed methods"**
> >
> > **A:** This is a very good question and also a very challenging one. Although our optimization converges and the losses decrease in Gaussianization, which indicates the decrease of KL-divergence, it is hard to measure Gaussianity in the high-dimensional space. We currently only demonstrate the effectiveness indirectly from inversion results.
> >
> > ---
> >
> >
> > > **2. "Many works ignore that MRI measurements are complex, not real, and this work also seems to make the same mistake. More realistic simulations for MRI including multi-coils (all modern MRI scanners are using multi-coils!) and complex Gaussian noise."**
> >
> > **A:** Thank you for pointing this out. In our background information on MRI (App. A1, Eqn. 13), we state that the data are complex and include the multi-coil formulation. The Gaussian noise that we added in our MRI experiments is complex. We will make it clear in our revised version. We only use the single-coil setup to compare with prior works, which usually use the same configuration. Also, our contribution focuses on deep generative models. An expert in medical imaging can easily extend our work to multi-coils.
> >
> > ---
> >
> > > **3. "Glow is an important work, but it seems like an outdated method as a compared method and there are a number of related recent works with better performance. Comparing with more recent methods will make this work stronger. For StyleGANv2, comparing with (Wulff & Torralba 2020) will help this work to be demonstrated as a SOTA method. Comparing with score matching based works will be surely a plus."**
> >
> > **A:** As mentioned previously, we have compared with a StyleGANv2 inversion using Wulff & Torralba 2020 and demonstrated the advantages of our method. In the revised version, we have included another motivating example using **Stable Diffusion** and demonstrated the necessity of our layers for potential applications such as "language-guided inversion." Please see this [*thread*](https://openreview.net/forum?id=OXP9Ns0gnIq&noteId=ueizSMYw4B) for our discussion on scored-based methods.
> >
> > ---
> >
> > >**4. "Gaussianization technique is also used in other image generation works. See Yen-Chi Cheng et al., InOut : Diverse Image Outpainting via GAN Inversion, CVPR 2022. This work should survey various image generation works with Gaussianization not only for compressive sensing MRI, deblurring and eikonal tomography, but also for other related inverse problems."**
> >
> > **A:** Yen-Chi Cheng et al. uses the 1D Gaussianization method proposed in Wulff & Torralba 2020, which we already compared in the paper. We positioned our paper for scientific applications, so we only mentioned those three typical inverse problems. We will cite Yen-Chi Cheng et al. and consider mentioning other related inverse problems, such as inpainting and outpainting.

---

### Official Review · Reviewer_YtuR · 2022-10-28

**Confidence:** 3
**Correctness:** 4
**Technical Novelty And Significance:** 3
**Empirical Novelty And Significance:** 3
**Recommendation:** 8

**Clarity, Quality, Novelty And Reproducibility:**

The paper is well organized. The method is novel and the experimental results show that the method is promising.

**Strength And Weaknesses:**

Strong points:
The authors point out that when the latent variables of deep generative models deviate from the desired Gaussian distribution, the solutions of inverse problems would be poor, and propose a method that makes the latent variables obey a Gaussian distribution. Using ICA, Yeo-Johnsonn transformation, and Lambert WxFx transformationo, the methd improves the independency, skewness, and heavy-tailedness of the distributions. The method is novel and the experimental results show that the proposed method can constrain the solutions more correctly.

Weak points:
The algorithm for the update of v is not clearly described in the text. Appendix F describes the gradient computation of the optimization-based differentiable layers but the reviewer believes the contents shown in Appendix F should be described in the text.

**Summary Of The Paper:**

The paper improves the deep generative models used for constraining solution spaces in inverse problems by making distributions of latent variables obey a Gaussian. Deep generative models such as StyleGAN2 and Glow are widely employed for the regularizes in inverse problems. Given latent variables that obey a Gaussian, these generative models generate realistic images. The authors point out that the latent variables deviate from the desired Gaussian distribution during inversion lead to poor results, and the proposed method reparameterizes and Gaussianizes the latent variables during the optimization for solving inverse problems. The proposed method was applied to compressive sensing MRI, image deblurring and eikoonal traveltime tomography, and the experimental results show the proposed method works well.

**Summary Of The Review:**

The reviewer recommends to accept this paper because of the strong points described above.

---

> ### Author Response · Authors · 2022-11-18
> **Response to Reviewer YtuR**
>
> Thank you very much for your review and recommendation!
> > **"The algorithm for the update of v is not clearly described in the text. Appendix F describes the gradient computation of the optimization-based differentiable layers but the reviewer believes the contents shown in Appendix F should be described in the text.""**
>
> **A:** We have added the suggested information in the caption of Figure 2. "Gradient computation in the Gaussianization layers is enabled by the implicit function theorem and automatic differentiation (App. G). We use the L-BFGS optimizer (Nocedel & Wright, 2006) to update $\mathbf{v}$.

---

### Official Review · Reviewer_VNJA · 2022-10-31

**Confidence:** 4
**Correctness:** 3
**Technical Novelty And Significance:** 3
**Empirical Novelty And Significance:** 3
**Recommendation:** 6

**Clarity, Quality, Novelty And Reproducibility:**

**Clarity**:

I found the paper to be quite clear, but perhaps a little verbose. An algorithm environment that describes the final updates, as well as inputs, parameters, and outputs, would be very useful. The information is very spread out in the current format.

Another nitpick is that the authors use the term ``Gaussianization'' to sometimes refer to Step 2 (whitening), and sometimes to refer to both steps of ICA + Whitening. Some consistency would be useful to avoid confusion about what sub-component of the algorithm is being described.

**Quality and originality**:

I think the paper meets the standard for ICLR. The proposed solution is sufficiently novel, and the problem of inverse problems using generative models has significant practical and theoretical significance.

**Strength And Weaknesses:**

**Strengths**:

- I think the proposed solution is very practically reasonable. The algorithm requires minimal ``training'' to force Gaussian statistics on the solution. Previous works have considered using the vanilla $\ell_2$ norm of $z$ (as in Bora et al 2017), while algorithms like PULSE[3] forcibly project to the Gaussian sphere. This solution is much better.

- The decomposition in Eqn 6 is quite clever, as is the algorithm for iteratively minimizing the two terms in the decomposition -- the ICA keeps the LHS of Eqn 6 constant and increases the second term in the RHS, which implies the first term in the RHS must decrease; the next step is a whitening step that forces the second term to decrease while leaving the first term constant.

- The experiments look really good, but it's hard to judge because the baselines considered are not the strongest.

**Weaknesses**:

- The motivation behind this problem is that an optimization problem as in Eqn (3) is unstable to choice of $\beta$, if the regularizer is $\beta || z ||^2$, as $z$ is supposed to be Gaussian distributed. But this doesn't make much sense in theory, as eqn (3) solves for the MAP in $z$, [which is not the same as searching for the MAP in $g(z)$ space, since $- \log p(G(z)) = ||z||^2 - \log det( \nabla_z G(z) ) $]. This inconsistency in performing MAP in $G(z)$ or $z$ space could be the reason for the artefacts, as opposed to issues in optimization / distribution of the latent vectors.

- An additional point about MAP in $z$ space: the phenomena described in this paper can also be circumvented when using GLOW by using something like Langevin dynamics to approximate Posterior Sampling [1]. In the case of StyleGAN, score-guided optimization[2] has been useful. In both cases the statistics of $z$ are respected. Your argument would be much stronger if you can offer a direct advantage (methodologically or experimentally) over these techniques. Does your optimization offer advantages over these?

- The related work section is not very accurate -- score models can be used for non-linear inverse problems as well. The cited references [Jalal et al, Song et al] considered compressed sensing MRI and CT, which are linear inverse problems. I am not familiar with work questioning their applicability to non-linear inverse problems.

- The baselines used for comparison are very weak, especially in Fig 4. To clarify, I'm not blaming the authors, it's well known that inverting a StyleGAN is empirically challenging. Comparing to compressed sensing experiments in [2] on the celebA dataset + StyleGAN would be a much more convincing and necessary experiment. (code and models: https://github.com/giannisdaras/sgilo )

- The optimization procedure makes sense, but is there any guarantee that ICA and whitening leads to a strict decrease? I don't know what the theoretical guarantees for ICA are, and it would be helpful if you could add some references in the appendix.

- Figure 5 has clear artefacts for the proposed method. What's going on? Can you provide a comparison to Langevin dynamics[1] on the same experiment?

**Clarifications**:

- The experiments section is a little confusing -- why do you have separate experiments for the Orthogonal Reparameterization and Gaussianization layers? I thought they're two components of one algorithm? What are these things in relation to Figure 2?

-

**References**:

[1] Jalal, Ajil, Sushrut Karmalkar, Alex Dimakis, and Eric Price. "Instance-Optimal Compressed Sensing via Posterior Sampling." In International Conference on Machine Learning, pp. 4709-4720. PMLR, 2021.

[2] Daras, Giannis, Yuval Dagan, Alex Dimakis, and Constantinos Daskalakis. "Score-Guided Intermediate Level Optimization: Fast Langevin Mixing for Inverse Problems." In International Conference on Machine Learning, pp. 4722-4753. PMLR, 2022.

[3] Menon, Sachit, Alexandru Damian, Shijia Hu, Nikhil Ravi, and Cynthia Rudin. "Pulse: Self-supervised photo upsampling via latent space exploration of generative models." In Proceedings of the ieee/cvf conference on computer vision and pattern recognition, pp. 2437-2445. 2020.

**Summary Of The Paper:**

This paper tries solving inverse problems using generative models as regularizers. It is known that when doing MAP / Maximum Likelihood optimization, the optimization variable deviates from its expected statistics, which is standard Gaussian in most cases.

This paper proposes an algorithm that can enforce Gaussian statistics on the latent variable $z$ during optimization. This is done by adding a KL divergence regularizer to the optimization objective, where the KL is between the distribution of $z$ and a standard Gaussian. The authors then decompose the KL divergence as a sum of two non-negative terms, and minimization is done by alternating ICA / whitening on the two terms in the decomposition.

The algorithm is novel and the experiments are interesting, although I have some concerns about the baselines used for comparison.

**Summary Of The Review:**

The algorithm is novel. The experiments are sort of convincing, but have clear issues. I am happy to raise my score if the authors can compare to the additional baselines I mentioned.

---

> ### Author Response · Authors · 2022-11-16
> **Response to Reviewer VNJA (Part 1 - on Glow + Langevin dynamics)**
>
> Thank you very much for the thought-provoking comments and suggestions! Let us first respond to your suggestions on comparing Langevin dynamics using Glow and SGILO.
>
> The Glow + Langevin dynamics results and the comparison with our method can be found in this [**LINK**](https://drive.google.com/file/d/1nak6DUYNiE3zD4VTo-CBF_t21xdPcERS/view?usp=sharing). As suggested, we ran experiments on the examples shown in Figure 5. We can see that the results from Langevin dynamics have a larger discrepancy from the ground truth than those from our method, and there are stronger artifacts. This preliminary result suggests that it is also challenging for Langevin dynamics to deal with noisy data.
> The Langevin dynamics we used were annealed Langevin dynamics as in [1] and  [2]:
> $$
> \boldsymbol{z}\_t=\boldsymbol{z}\_{t-1}+\frac{\alpha\_i}{2} \nabla\_{\boldsymbol{z}} \log p\left(\boldsymbol{z}\_{t-1} | \boldsymbol{d}\right)+\sqrt{\alpha\_i} \boldsymbol{\eta}\_t,
> $$
> where $\boldsymbol{\eta}\_t \sim \mathcal{N}(0,I)$, and the step size $\alpha\_i = \epsilon \cdot \sigma\_i^2/\sigma\_L^2$ was constructed in a similar way as in [1] and [2]: $\\{\sigma\_i\\}_{i=1}^{L}$ is a geometric sequence. We chose $\sigma\_1 = 10$, $\sigma\_{10}=0.1$, and $L=10$. At each noise level, we ran 200 steps. The fixed step size $\epsilon$ was chosen as 1e-9 because we found that in the delubrring experiment if it is larger than 1e-8 the results would blow up.
>
> Besides, to compute $\nabla\_{\mathbf{z}} \log p\left(\mathbf{z}\_{t} | \mathbf{d}\right)$, one has to assume a standard deviation of noise $\sigma$. We find this hyper-parameter very challenging to pick, and it dramatically affects inversion results. After some trial and error, we empirically picked $\sigma=10$. This may not be the best choice, and one might be able to improve it with more careful picking or annealing of the likelihood term. But the observation highlights the advantage of our method: it does not need such complicated hyper-parameter tuning.
>
>
>
> [1] Jalal, Ajil, Sushrut Karmalkar, Alex Dimakis, and Eric Price. "Instance-Optimal Compressed Sensing via Posterior Sampling." In International Conference on Machine Learning, pp. 4709-4720. PMLR, 2021.
>
> [2] Song, Yang, Liyue Shen, Lei Xing, and Stefano Ermon. "Solving inverse problems in medical imaging with score-based generative models." arXiv preprint arXiv:2111.08005 (2021).

---

> > ### Author Response · Authors · 2022-11-16
> > **Response to Reviewer VNJA (Part 2 - on SGILO and scored-based models)**
> >
> > Thank you for bringing our attention to this nice paper about SGILO! It is a very interesting contribution that introduces the idea of score guidance into general DGM-regularized inversion. We will cite it in our revised version. Before our discussion, we would like to mention that SGILO was just published in ICML 2022, only around two months before the ICLR submission deadline. It is considered contemporaneous (not required to be compared with) according to ICLR guidelines. Due to certain constraints, we are unable to conduct thorough experiments to compare it properly with our method at this moment. We hope to have the reviewer's kind understanding.
> >
> > On the other hand, the main selling point of our method is its ability to deal with noisy data and handle nonlinear inverse problems and that it can be applied to any DGMs with a latent space. Therefore, the experiment proposed by the reviewer is to answer this core question: is the score-based method also able to deal with noisy data and nonlinear problems? A comparison with SGILO will not give us a direct answer since SGILO has many additional components. For example, it requires using ILO to invert a dataset first to get intermediate layer outputs to form an additional training dataset to train a vision transformer for score guidance.
> >
> > Interestingly, we find another submission to ICLR2023 that directly addresses this question.
> >
> > **Title**: DIFFUSION POSTERIOR SAMPLING FOR GENERAL NOISY INVERSE PROBLEMS
> >
> > **Link**: https://openreview.net/forum?id=OnD9zGAGT0k
> >
> > This paper finds that previous scored-based inversion methods (e.g., Jalal et al., 2021 and Song et al., 2021) (1) cannot handle measurement noise, and (2) "either fails or is not straightforward" to solve nonlinear inverse problems. Then, this paper provides its solution. Another ICLR2023 submission on the same topic can be found here.
> >
> > **Title**: PSEUDOINVERSE-GUIDED DIFFUSION MODELS FOR INVERSE PROBLEMS
> >
> > **Link**: https://openreview.net/forum?id=9_gsMA8MRKQ
> >
> > They both seem to be the first to handle both measurement noise and nonlinear forward models for inverse problems solved using scored-based models. Based on these papers, score-based models are not immune to data noise and cannot handle nonlinearity naturally. It is an active research direction.
> >
> > Besides, scored-based methods might not work for certain physics-based (multi-parameter) inverse problems. In such problems, starting with Gaussian noise as model parameters may break the physics simulation engine. For example, in elastic waveform inversion, initializing material properties using Gaussian noise may create unrealistic scenarios where the P-wave speed is lower than the S-wave speed. Our method does not have this problem.

---

> > > ### Author Response · Authors · 2022-11-16
> > > **Response to Reviewer VNJA (Part 3 - other questions)**
> > >
> > > Here is our response to your remaining questions and comments.
> > >
> > > > **1. "...But this doesn't make much sense in theory, as eqn (3) solves for the MAP in z, which is not the same as searching for the MAP in g(z) space... This inconsistency in performing MAP in G(z) or z space could be the reason for the artefacts, as opposed to issues in optimization / distribution of the latent vectors."**
> > >
> > > **A:** Thank you for pointing this out. We are also aware of this issue. Therefore, the motivating results shown in Fig. 20 are obtained by finding the MAP estimate in the parameter space rather than in the z space. Please see our Glow-regularized inversion formulation in Appendix J1. On the other hand, an MAP point may not be well defined in the DGM. As for GAN, we are just arguing that the traditional formulation of only penalizing the norm of z may not be effective in reducing artifacts.
> > >
> > > ---
> > > > 2. **"The related work section is not very accurate -- score models can be used for non-linear inverse problems as well. The cited references [Jalal et al, Song et al] considered compressed sensing MRI and CT, which are linear inverse problems. I am not familiar with work questioning their applicability to non-linear inverse problems."**
> > >
> > > **A:** Thank you for the comment. In Song et al., the derivation of the conditional stochastic process $\\{ \mathbf{y}\_t | \mathbf{y}\\}\_{t \in [0,1] }$ assumes that the model is linear (page 5, the second paragraph under section 3.2). In Jalal et al., they only tested it on linear problems. It is not clear what the performance is like on nonlinear problems. Nonetheless, to make our statement more accurate, we will change our original words to something like "They have been mainly applied to linear inverse problems."
> > >
> > > ---
> > > > 3. **"The optimization procedure makes sense, but is there any guarantee that ICA and whitening leads to a strict decrease? I don't know what the theoretical guarantees for ICA are, and it would be helpful if you could add some references in the appendix."**
> > >
> > > **A:** We cited the paper [1] that proves the convergence of ICA at the end of Appendix 3.1.
> > >
> > > [1] Oja, Erkki, and Zhijian Yuan. "The FastICA algorithm revisited: Convergence analysis." IEEE transactions on Neural Networks 17, no. 6 (2006): 1370-1381.

---

> > > > ### Author Response · Authors · 2022-11-16
> > > > **Response to Reviewer VNJA (Part 4 -- other questions)**
> > > >
> > > > > 4. **"Figure 5 has clear artefacts for the proposed method. What's going on? Can you provide a comparison to Langevin dynamics on the same experiment?"**
> > > >
> > > > **A:** We have provided a comparison to Langevin dynamics in a previous post. As for the artifacts, we think they may be caused by some correlation between neighboring patches. After all, our method is a necessary condition of the Gaussian prior. The joint distribution of random tensors (patches) may not be Gaussian. One can further improve the performance using the rolling operation (App. J) and additional Gaussianization layers, with increased computational cost.
> > > >
> > > > ---
> > > > > 5. **"The experiments section is a little confusing -- why do you have separate experiments for the Orthogonal Reparameterization and Gaussianization layers? I thought they're two components of one algorithm? What are these things in relation to Figure 2?"**
> > > >
> > > > **A:** Sorry for the confusion. The Orthogonal Reparameterization is actually a different algorithm that we came up with to compare with the Gaussianization layers. The details of it are in Appendix H. The idea is to fix the random vector $\mathbf{z}$ with a parameterized orthogonal matrix $R$ and a fixed standard Gaussian vector $\mathbf{v}$. No matter how the inversion changes $R$, $\mathbf{z}$ should be kept Gaussian. However, the orthogonal reparameterization generally underperforms the Gaussianization layers. Our interpretation is as follows: rather than actively destroying latent tensor patterns as the Gaussianization layers do, orthogonal reparameterization cannot prevent permuting the ﬁxed typical Gaussian latent tensors into ones with spatial patterns (i.e., out of the typical set of a Gaussian).
> > > >
> > > > ---
> > > > >6. **"An algorithm environment that describes the final updates, as well as inputs, parameters, and outputs, would be very useful."**
> > > >
> > > > **A:** Thank you for this suggestion! We also thought about creating an algorithm like this. However, we think it might be better to use the diagram in Figure 2 to illustrate the inputs and outputs because we already have several algorithm environments, and we are afraid that we might confuse our readers by introducing another one. Also, it is now very challenging to fit an algorithm environment in the main text due to the page limit.
> > > >
> > > > ---
> > > > > 7. **"Another nitpick is that the authors use the term ``Gaussianization'' to sometimes refer to Step 2 (whitening), and sometimes to refer to both steps of ICA + Whitening. Some consistency would be useful to avoid confusion about what sub-component of the algorithm is being described."**
> > > >
> > > > **A:** Thank you for pointing this out. It can be confusing since whitening and ICA are used together. We will use "whitening only" to refer to cases where no subsequent ICA component exists.

---

### Official Review · Reviewer_NCs6 · 2022-11-03

**Confidence:** 3
**Correctness:** 4
**Technical Novelty And Significance:** 3
**Empirical Novelty And Significance:** 3
**Recommendation:** 6

**Clarity, Quality, Novelty And Reproducibility:**

Clarity: OK. It could be improved (see comments above).
Novelty: The contribution seems novel. However, a version of this manuscript has been available on arXiv since Dec 2021. Can the authors comment on this?
Reproducibility: The authors should realise code if the manuscript is accepted.

**Strength And Weaknesses:**

Strengths:
* The new Gaussianization layer is a novel technical contribution, which combines several sublayers (ICA layer, power transform layer and lambert layer). This layer and its component can be applied to various deep generative models, inverse problems and deep variational inference models, thus being of potential interest to a broad community.
* Results are convincing. Additional experiment/results are provided in the supplement. The authors also considered an ablation study to investigate which parts of the proposed Gaussianization layers were responsible for the observed performance improvement.

Weakness:
* I think the presentation of the paper could be improved. The authors may consider discussing a specific example when introducing inverse problems, regularization with generative models and the gaussianization layer. The deblurring example would be the easiest for this (the authors may also consider discussing this as the first experiment for consistency).
* The authors do not comment on (future) code availability. I think that including Pytorch/Tensorflow implementations of the proposed Gaussian layer and its subparts would make this work more impactful. If the paper is accepted, the code to reproduce experiments should also be released.


**Summary Of The Paper:**

Recent work has shown how deep generative models (such as GANs or VAEs) can be used to regularize inverse problems by enabling optimization in the latent space of a pre-trained generative model—with a regularisation term to encourage latent code gaussianity. However, low-fidelity solutions whose latent codes strongly deviate from gaussian still occur. To address this issue, the authors propose a new approach: (i) perform optimization on unconstrained latent code, (ii) leverage a novel gaussianization layer to map the unconstrained latents to gaussian ones, which are then input to the generative model.

The authors apply their approach to three different inverse problems (compressive-sensing MRI, image deblurring, and eikonal tomography) and compare their approach to different methods for inverse problems (including methods that use generative models but different strategies to enforce gaussian latent codes). Across the three applications, the proposed approach achieves state-of-the-art performances.

**Summary Of The Review:**

The authors introduce a new strategy for solving inverse problems using generative models by introducing a new layer enabling latent code gaussianization. This is potentially an important contribution for inverse problems, generative models and deep variational inference. The authors could improve paper readability providing a guiding example (see comments above) and release code for the proposed layers for broader impact. If the paper is accepted, the authors should provide code to reproduce their experiments.

---

> ### Author Response · Authors · 2022-11-05
> **Response to Reviewer NCs6**
>
> Thank you very much for your comments and suggestions! Please find our response below.
>
> > **1 ."The authors may consider discussing a specific example when introducing inverse problems, regularization with generative models and the gaussianization layer. The deblurring example would be the easiest for this (the authors may also consider discussing this as the first experiment for consistency).**
>
> **A:** Thank you for this suggestion! Indeed, a specific example in the introduction can be very helpful for understanding. But we are afraid that this major rewrite will exceed the page limit. Also, we would like to make our formulation as general as possible and emphasize its applications in scientific problems. Based on our previous submission experience, a deblurring example at the beginning may bias readers towards this subfield of CV. To provide the necessary background information, we point readers to Appendix A for the details of inverse problems in the second paragraph of the Introduction.
>
> ---
>
> > **2. "The authors do not comment on (future) code availability. I think that including Pytorch/Tensorflow implementations of the proposed Gaussian layer and its subparts would make this work more impactful. If the paper is accepted, the code to reproduce experiments should also be released."**
>
> **A:** We would like to share our code with reviewers using an anonymous link in another post. The implementation is in PyTorch. We will definitely share the code with the public and continue improving it after the paper is accepted.
>
> ---
>
> > **3. "Novelty: The contribution seems novel. However, a version of this manuscript has been available on arXiv since Dec 2021. Can the authors comment on this?"**
>
> **A:** To remain anonymous, we cannot link our identity to arXiv papers. We can only say we first submitted our paper last November, and it went through review cycles. The quality of the paper has been greatly improved thanks to the reviewers' valuable feedback.

---

### Author Response · Authors · 2022-11-18
**Paper revision**

We would like to thank all reviewers for their great feedback. Please find below the edits to our paper.

1. We added motivating examples about Stable Diffusion. We created a standalone section: ADDITIONAL MOTIVATING EXAMPLES in Appendix B.

2. We included SGILO and InOut in the references.

3. We cited Meng et al.

3. In the caption of Figure 2, we added information about gradient computation and how to update $\mathbf{v}$.

4. In Related Work, we modified our description of scored-based methods. It now reads, "they have been mainly applied to linear inverse problems and challenged by noisy data (Kawar et al., 2021)." We also added some potential limitations of scored-based methods in solving certain physics-based inverse problems.

5. In section 4.1, we clarified that the Gaussian noise is complex.

6. We use "whitening only" or "whitening alone" to refer to cases where there is no ICA afterward.

---

### Decision · Program_Chairs · 2023-01-20

**Decision:**

Accept: poster

**Justification For Why Not Higher Score:**

The Gaussianization idea is useful, but it's only useful for a relatively narrow set of applications and methods. I, therefore, don't think that the paper should be highlighted at the conference.

**Justification For Why Not Lower Score:**

See meta review.

**Metareview: Summary, Strengths And Weaknesses:**

The paper considers the problem of solving inverse problems by imposing a generative prior. Many generative priors that are used in practice (GANs, Flow-based models) typically have a Gaussian latent representation. The paper argues that by imposing a generative model, the identified latent representation can deviate from a Gaussian distribution, which can lead to sub-par results. To address this issue, the paper introduces layers to Gaussianize the latent tensor. The paper then studies three inverse problems (accelerated MRI, deblurring, and tomography) and shows that the method works well.

The proposed Gaussianization approach is a new and potentially useful idea for solving inverse problems with generative priors, and the paper shows that the method works well on three inverse problems, and for different generative models. Previous approaches have also proposed or implicitly adopted ``Gaussianization'' approaches, such as including an l2 norm and projecting onto the Gaussian sphere, but the approach provided in the paper performs well, and is a useful idea.

The paper was discussed in detail among the reviewers, and the reviewers and I find that this is a valuable contribution to the literature of signal reconstruction with generative priors. If the authors can manage, it would be valuable to compare to the recently proposed method:
[2] Daras, Giannis, Yuval Dagan, Alex Dimakis, and Constantinos Daskalakis. "Score-Guided Intermediate Level Optimization: Fast Langevin Mixing for Inverse Problems." In International Conference on Machine Learning, pp. 4722-4753. PMLR, 2022.
as suggested by R2.



**Note From Pc:**

if the above contains the word "oral" or "spotlight" please see: "oral" presentation means -> notable-top-5% and "spotlight" means -> notable-top-25%. As stated in our emails, we are disassociating presentation type from AC recommendations

**Summary Of Ac-Reviewer Meeting:**

After the discussion (virtual and online) three out of the four reviewers are supportive of publication.

R1 argues for accepting the paper (6), since the methods is potentially useful.

R2 (6) notes that the idea is good, and from the experiments, the method also looks good. However, the reviewer mentioned that the authors could have chosen stronger baselines. After the rebuttal and discussion, this is not a major concern, and the reviewer is supportive of publication.

R3 (8): Supportive, no major issues. The reviewer provided a detailed account on why the method is sufficiently different from the Meng et al. paper (similarity to that paper has been brought up by R4), which was very helpful for the discussion.

R4 (initially 3, now 5): Initially finds that the main issue is that the Gaussianization idea is close to Meng et al's work. The authors respond that they actually cited Meng et al and were inspired by that paper, but then removed the citation, and will now add it again. This was discussed during the rebuttal, and the reviewers agree that the method is sufficiently different from Meng et al's work. The reviewer increased the score to 5 during the rebuttal period.

R4 notes that minor issues remain like somewhat simplistic problem settings (single vs multi-coil in MRI). I don't think this is a major issue, as we would not expect the comparative results to be significantly different when moving from single to multi-coil.